# On the relation between avalanche occurrence and avalanche danger level

Jürg Schweizer[1], Christoph Mitterer[2], Frank Techel[1], Andreas Stoffel[1], Benjamin Reuter[1,3]

[1] WSL Institute for Snow and Avalanche Research SLF, Davos, Switzerland
[2] Avalanche Forecasting Service Tyrol, Innsbruck, Austria
[3] Météo France, CNRS, CNRM, Centre d'Étude de la Neige, Grenoble, France

*Correspondence to*: Jürg Schweizer (schweizer@slf.ch)

**Abstract.** In many countries with seasonally snow-covered mountain ranges warnings are issued to alert the public about imminent avalanche danger, mostly employing an ordinal, five-level danger scale. However, as avalanche danger cannot be measured, the characterization of avalanche danger remains qualitative. The probability of avalanche occurrence in combination with the expected avalanche type and size decide on the degree of danger in a given forecast region ($\gtrsim 100$ km$^2$). To

describe avalanche occurrence probability the snowpack stability and its spatial distribution need to be assessed. To quantify the relation between avalanche occurrence and avalanche danger level we analyzed a large data set of visually observed avalanches (13'918 in total) from the region of Davos (Eastern Swiss Alps, ~ 300 km$^2$), all with mapped outlines, and compared the avalanche activity to the forecast danger level on the day of occurrence (3533 danger ratings). The number of avalanches per day strongly increased with increasing danger level confirming that not only the release probability but also the frequency

of locations with a weakness in the snowpack where avalanches may initiate from, increases within a region. Avalanche size did generally not increase with increasing avalanche danger level, suggesting that avalanche size may be of secondary importance compared to snowpack stability and its distribution when assessing the danger level. Moreover, the frequency of wet-snow avalanches was found to be higher than the frequency of dry-snow avalanches for a given day and danger level; also, wet-snow avalanches tended to be larger. This finding may indicate that the danger scale is not used consistently with regard

to avalanche type. Although, observed avalanche occurrence and avalanche danger level are subject to uncertainties, our findings on the characteristics of avalanche activity suggest reworking the definitions of the European avalanche danger scale. The description of the danger levels can be improved, in particular by quantifying some of the many proportional quantifiers. For instance, based on our analyses 'many avalanches', expected at danger level *4–High,* means on the order of at least 10 avalanches per 100 km$^2$. Whereas our data set is one of the most comprehensive, visually observed avalanche records are known

to be inherently incomplete so that our results often refer to a lower limit and should be confirmed using other similarly comprehensive data sets.

# 1    Introduction

Avalanche forecasting was described by McClung (2002) as the prediction of snow instability in space and time relative to a given triggering level. The main sources of uncertainty in forecasting are the unknown temporal evolution and the spatial variations of instability in the snow cover. For these reasons predictability of snow avalanche occurrence is limited; it is inversely related to scale, i.e. a probability of occurrence can be given at the regional scale, but not at the scale of a single avalanche path (Schweizer, 2008). In forecasting of natural systems, in which variations may or may not be random, a distinction is often made between forecasting and prediction. In our case, prediction means precisely defining when and where an avalanche occurs. Forecasting, on the other hand, implies describing the probability of avalanche occurrence within a certain time frame and area. Given these definitions it is obvious that prediction is not possible – even though it would be desirable – whereas forecasting is certainly possible but inherently includes uncertainty as the forecast is probabilistic (Silver, 2012).

Even if avalanche forecasting is probabilistic and includes uncertainty, it should be grounded in clear definitions and uncertainty should not stem from ambiguous definitions but the nature of the problem. In public forecasting, i.e. issuing bulletins describing the avalanche situation, avalanche hazard is described by one of five avalanche danger levels. The danger levels (*1–Low, 2–Moderate, 3–Considerable, 4–High, 5–Very High*) are defined in the avalanche danger scale that was originally agreed by the European avalanche warning services in 1993 (EAWS, 2019a; Meister, 1995). Subsequently, a very similar five-level scale was adopted in North America (Dennis and Moore, 1997), which was later revised with an emphasis on risk communication (Statham et al., 2010). In the original European danger scale, the avalanche danger levels were defined in terms of the release (or triggering) probability, the frequency and location of triggering spots and the potential avalanche size. All three elements are supposed to be combined when assigning a danger level to a given avalanche situation. Moreover, it is assumed that all three elements increase with increasing avalanche hazard. However, the definitions for the different danger levels are short, qualitative descriptions and leave room for widely varying interpretations (Müller et al., 2016a). Not surprisingly, a recent study that looked at forecast differences across borders of contiguous forecast areas suggests that remarkable inconsistencies in the application of the danger levels exist (Techel et al., 2018). Based on a survey among forecasters, Lazar et al. (2016) also found substantial differences in assigning a single danger rating to a given scenario of avalanche conditions. These studies demonstrate that there is a lack of quantification with regard to the three key elements and their links in the avalanche danger scale.

This lack of formal underpinnings, among other reasons, motivated the development of a conceptual model of avalanche hazard in North America, which essentially formalizes the hazard assessment process (Statham et al., 2018). However, the final step on how to derive the danger level is not described. In Europe, the so-called Bavarian matrix was developed to support the decision process in forecasting. It is basically a look-up-table that allows assigning the danger level based on the probability of avalanche release and the frequency of triggering spots (Müller et al., 2016a). Avalanche size is not explicitly considered in the Bavarian matrix. Hence, recent developments in Europe were aiming at including avalanche size and harmonizing the

European with the North American approach. To this end, an approach with two matrices, a so-called likelihood matrix and a danger matrix, was suggested in an attempt to merge the concepts behind the conceptual model of avalanche hazard with the Bavarian matrix (Müller et al., 2016a; Müller et al., 2016b). Also, a version of the Bavarian matrix including avalanche size was suggested (EAWS, 2020).

There are few data-driven studies that link the avalanche danger level to any of the three key elements. Haegeli et al. (2012) analyzed two years of public avalanche forecasts with underlying hazard assessments by Avalanche Canada. They found that the maximum likelihood of triggering had the strongest impact on danger rating selection; the second most important predictor variable was the maximum expected avalanche size. More recent analyses on the relation between the components of the conceptual model of avalanche hazard and the danger ratings showed that identical avalanche scenarios were often rated differently – possibly indicating substantial inconsistencies in the forecasting process. This finding is likely due to the lack of explicitly assigning danger ratings to the various combinations in the likelihood-magnitude chart (Clark and Haegeli, 2018; Clark, 2019).

The avalanche danger levels can also be characterized with observational data related to snow instability. In the context of a verification campaign, Schweizer et al. (2003) established typical stability distributions for the danger levels *1–Low*, *2–Moderate* and *3–Considerable* based on many snow instability tests for single avalanche situations. Likewise, signs of instability such as whumpfs, shooting cracks and recent avalanching were related to the danger levels (Jamieson et al., 2009a; Schweizer, 2010). As shooting cracks were almost ten times more frequent at *3–Considerable* (or higher) than at *2–Moderate* (or lower), they had most predicting power in a simple classification tree. Avalanche activity was only considered as binary variable, which does not allow insight into the avalanche characteristics at a given danger level.

Given the lack of quantitative definitions in the avalanche danger scale, our aim is to characterize avalanche activity with regard to avalanche hazard. We therefore analyzed a large data set of avalanche observations from the region of Davos (Eastern Swiss Alps) and compared the avalanche activity to the avalanche danger forecast. Though both variables are subject to uncertainty, we aim at characterizing the danger levels based on frequency, type and size of avalanche occurrence.

## 2  Data and methods

We analyzed a 21-year data set of manually observed avalanche occurrences from the region of Davos, an area of about 300 km$^2$. Large parts of the study area ranging between 1200 and 3200 m a.s.l. are steep mountain terrain. According to the avalanche terrain classification (CAT) by Harvey et al. (2018) about 67 % of our study area are considered avalanche terrain. The snow climate in the region of Davos can be described as transitional (McClung and Schaerer, 2006).

Data cover the winters from 1998-1999 to 2018-2019 and include 13,918 individual avalanches, which were all mapped (Figure A in the Supplement). Avalanches were recorded on a daily basis by SLF as well as by ski resort staff. In the more remote

parts of the study area the observations are in general less consistent. Also, during the first six years, fewer avalanches were recorded than during the rest of the study period.

For each avalanche, we derived avalanche length $L$ and width $W$ from a rectangle of the smallest width enclosing the mapped perimeter using the 'minimum boundary geometry' tool in ArcGIS. Based on these values of avalanche length and width we assigned the avalanche size class (1 to 4) according to the Canadian size classification (McClung and Schaerer, 2006). Since avalanches of size class 5 were rare ($L > 2000$ m and $W > 300$ m; $N = 11$), we assigned those to class 4. Moreover, 116 avalanches were too small to derive meaningful values of length and width, but were still assigned an avalanche size of 1 (hence $N = 13,802$ in Table 1 and Figure 1). Table 1 describes the criteria for size classification. Also given are the resulting median length, width and area per size class for our data set.

Figure 1 shows that our size classification based on mapped outlines well reproduced the exponential increase that underlies the original proposal for the size classification (McClung and Schaerer, 1981). They suggested classifying avalanche size $S$ based on mass and proposed five classes where $S = \log M$ with $M$ the avalanche mass given in tens of tons. Their intention was to derive a classification that is based on destructive power, which in the end is related to volume or length. Figure 1 suggests that estimating avalanche size based on avalanche length seems indeed feasible.

Table 1: Definition of avalanche size based on length and width of avalanche. Resulting median length, width and area per size class ($N = 13,802$).

| Avalanche size | Class length (m) | Operator | Class width (m) | $N$ | Median length (m) | Median width (m) | Median area (m²) |
|---|---|---|---|---|---|---|---|
| 1: small | $< 50$ | AND | $< 50$ | 501 | 37 | 24 | 654 |
| 2: medium | $< 300$ | AND | $< 300$ | 9766 | 144 | 42 | 3989 |
| 3: large | [300, 999] | OR | [300, 999] | 3228 | 430 | 91 | 21,252 |
| 4: very large | $\geq 1000$ | OR | $\geq 1000$ | 307 | 1196 | 256 | 144,113 |

In addition, the avalanche records included information on the type of triggering (natural, person, explosives/snow grooming machine, unknown) and the type of snow conditions. The snow conditions relate to the liquid water content in the starting zone (dry, wet, mixed, unknown). Dry and wet refer to dry-snow and wet-snow avalanches, respectively, whereas mixed is less well defined and typically refers to avalanches with dry-snow conditions in the starting zone, but wet-snow conditions in the track or runout zone. Our records of avalanche observations cover 1358 individual days.

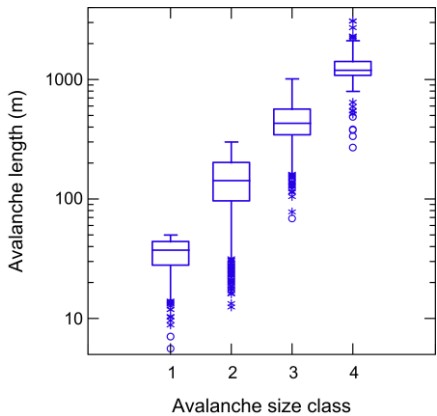

**Figure 1: Distribution of avalanche length per avalanche size class for the 21-year data set of avalanche observations from the region of Davos applying the classification criteria given in Table 1. Boxes span the interquartile range from 1st to 3rd quartile with a horizontal line showing the median. Whiskers show the range of observed values that fall within 1.5 times the interquartile range above the 3rd and below the 1st quartile. Asterisks and open circles refer to outliers and far-outliers beyond the fences. Numbers indicate avalanches per class; total number of avalanches: $N = 13,802$.**

We calculated the avalanche activity index (AAI) for each day using the usual weights for size classes 1 to 4, namely 0.01, 0.1, 1, and 10, respectively (Schweizer et al., 2003). Moreover, we considered the type of triggering using weights, namely 1 for natural avalanches, 0.5 for human-triggered avalanches, and 0.2 for the other artificially triggered avalanches (Föhn and Schweizer, 1995). For the avalanches with unknown trigger we assigned a weight of 0.81 since this was the weighted average of the triggering weight considering the frequency of avalanches for the three known triggering classes. In fact, almost all of the avalanches in the unknown triggering class were likely natural avalanches so that a weight of 0.81 was appropriate. We also calculated the individual AAI for the combinations of the various types of triggering and types of snow conditions.

We then merged the data set of avalanche observations with the avalanche danger as forecast in the public bulletin for that day and the region of Davos; the forecasting region is smaller than our study area, but is representative as it is located well in its center where most avalanches were recorded. For a total of 3533 days a danger rating was available. Some of the avalanches occurred outside the period when public forecasts were issued, e.g. in October or late May, and were not included for further analysis. This reduced the total number of observed avalanches to 13,745 and the number of days when at least one avalanche was recorded to 1301. This means, about every third day (37 %) with a danger rating at least one avalanche was observed for the 21-year period we analyzed.

As independent data to verify the issued danger level is not available for the entire data set, we compared the avalanche activity observed on a given day to the forecast danger level on that day. Nevertheless, we did some obvious data checking. For instance, at *4–High* we expect many natural avalanches. Indeed, the two highest danger levels can be verified by avalanche activity. Therefore, we scrutinized the forecast danger levels and adjusted them to most realistic values. All these corrections are summarized in the supplementary material in Table A1. We first checked the days with the highest danger levels, since in

these cases erroneous forecasts can most easily be detected: when the danger level was 4–*High* and no avalanches were observed, the forecast was either too high or the avalanche observations were not correctly assigned to the day of occurrence (but rather the day of observation).

We therefore started the correction process by checking the avalanche activity on the 51 days when the danger rating was either *4–High* or *5–Very High* (47 and 4 days, respectively). We found that on 30 out of 51 days the avalanche activity was zero or unusually low. For each of these days, we revisited the weather, snow and avalanche conditions in the relevant period. For 23 out of the 30 days we down-rated the danger. For the remaining 7 days we corrected a temporal mismatch between the date the hazard peaked and the date avalanches were registered. For example, occasionally all avalanche observations from a

3-day storm had been assigned to the first or last day of the storm. For 1 out of these 7 days we increased the danger level from *4–High* to *5–Very High*. This reduced the number of days with rating *4–High* from 47 to 25, and with rating *5–Very High* from 4 to 3. Still, on one day with danger rating *4–High* no avalanches were observed; this seems unlikely, but it was not possible to reconstruct the likely date of occurrence in that well-known storm period in February 1999. Unfortunately, records were in general rather inconsistent during the major storms in January and February 1999. For one day in January 2018, when the

forecast danger level was *4–High* and the avalanche activity very prominent (AAI = 158) a detailed verification revealed (Bründl et al., 2019) that the forecast danger level should have been *5–Very High*. This increased the total number of days with danger level *5–Very High* to 4. Hence, after these corrections, the danger rating was 5–*Very High* on 4 days, *4–High* on 25 days and *3–Considerable* on the remaining 22 days.

The median AAI of natural avalanches for the days with danger rating of either *4–High* or *5–Very High* was 13.6, which

corresponds to, for instance, only one avalanche of size 4 and a few smaller avalanches. Further quality checking revealed that there were a number of days with higher avalanche activity but lower danger level. In total on 59 days, the avalanche danger was rated *3–Considerable*, but many natural avalanches occurred. This was also the case for 17 days, when danger *2–Moderate* was forecast. Again, we checked all these cases against the weather, snow and avalanche conditions. For 57 of these 59 days we increased the rating from *3–Considerable* to *4–High* since the AAI clearly indicated that the avalanche activity had been

underestimated at the time of the forecast. On the remaining two days the number of natural avalanches was too low (<10) to justify a change. For 12 out of 17 days with forecast danger *2–Moderate*, we changed the danger level to *4–High* as many avalanches were observed, in most cases wet-snow avalanches, and the AAI was high. On the remaining 5 days we changed the danger level to *3–Considerable* as the total number of natural avalanches was too low (<10).

Subsequently, we considered the number of cases with *2–Moderate* danger, but an avalanche activity (only naturals) higher

than the median index (1.0) for days with *3–Considerable* danger. There were 99 days with AAI > 1.0. In 25 of these cases, the number of avalanches (size 2 and larger) was larger than 10. For these 27 days we changed the danger rating to *3–Considerable*. In 19 out of these 27 cases the avalanches were wet-snow avalanches. Finally, we adjusted the danger level from *1–Low* to *3–Considerable* for 2 days, one with high natural wet-snow avalanche activity and the other with several skier-triggered avalanches.

Overall, we changed 129 out of the 3533 danger ratings (3.7 %), mostly by one danger level, occasionally by two danger levels (12 %); in most cases (106 out of 129: 82 %) we increased the danger rating since there was clearly a rather high activity of natural avalanches. In total, there were finally 94 days (2.7 %) with danger rating *4–High*, still fairly few for 21 winter seasons.

For the analyses, we stratified the data into the danger levels and compared the avalanche activity index (AAI; Schweizer et al., 2003), the proportion of avalanche sizes, the proportion of days when avalanches were observed, and the average number
of avalanches per day.

The number of avalanches per day relates to the probability of avalanche occurrence in our study area, i.e. one metric integrating snow stability and its distribution. The lower stability is and the more frequent the triggering locations are, the more avalanches are observed.

To compare distributions, we used the non-parametric Mann-Whitney *U*-test. We selected a level of significance $p = 0.05$ to
judge whether the observed differences were significant. We also checked for equality of proportions in 2 x 2 contingency tables, for instance, do decide whether the proportion of size 2 avalanches was equal for two different danger levels. Relations of continuous data such as the AAI with ordinal variables were described with the Spearman rank-order correlation coefficient $r_s$ (Spiegel and Stephens, 1999).

## 3    Results

**3.1 Avalanche activity**

Figure 2 shows the avalanche activity index including all avalanches irrespective of triggering type or snow conditions and Table 2 summarizes some key figures on the avalanche activity with respect to danger level. The avalanche activity, expressed as AAI, increased with increasing danger level ($r_s = 0.42$, $p < 0.001$). The median values were 0.15, 0.21, 1, 21 and 88 for the avalanche danger levels 1 to 5. The increase was particular prominent from *3–Considerable* to *4–High*. The highest values
with AAI > 145 correspond to four distinct, well-known avalanche periods in the region of Davos: 23 April 2008, 9 March 2017, 22 January 2018 and 14 January 2019. There were only four days with danger level *5–Very High* so that the corresponding AAI statistics are indicative at best.

The proportion of days when avalanches were observed at a given danger level (AAI > 0), increased from about 9 % at *1–Low* to 99 % for *4–High* (Table 2). If natural avalanches were considered only, these proportions were 6 %, 16 %, 35 % and 95 %.
At *1–Low*, natural avalanches were observed at only 1 out of 16 days when this danger level was forecast. At *3–Considerable*, natural avalanches were recorded every third day and at *4–High* on almost all days. This increase of the proportion of avalanche days primarily reflects that snow stability decreases with increasing danger level.

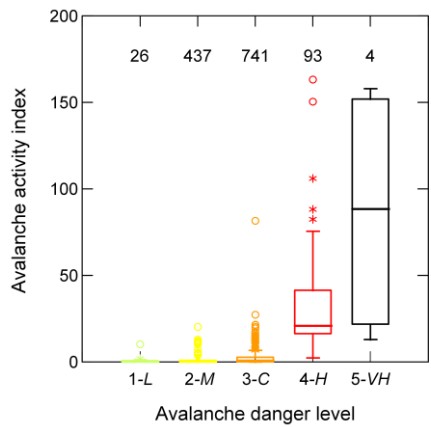

**Figure 2: Avalanche activity index AAI per danger level (*1–Low* to *5–Very High*). Only days with AAI > 0 are included. Numbers indicate number of days per danger level; total number of days: *N* = 1301.**

Moreover, the number of observed natural avalanches increased with increasing danger level. At the lower danger levels *1–Low* and *2–Moderate*, the median number of avalanches on a day with avalanche activity was 1; at *3–Considerable* the median number increased to 3, with an even stronger increase to 22 natural avalanches per day at *4–High*. As the avalanche records are likely incomplete for two out of four days with danger level *5–Very High* the median number of natural avalanches per day was only 19. This prominent increase of natural avalanche activity with increasing danger level is also evident in the AAI (only considering natural avalanches; median value): 0.1, 0.1, 0.2, 20 and 76 for the danger levels *1–Low* to *5–Very High*, respectively. The increase of the AAI or the number of avalanches per day reflects the increasing avalanche occurrence probability with increasing danger level. The prominent increase is due to decreasing snow stability and at the same time increasing frequency of locations with poor snow stability where avalanches can initiate from.

**Table 2: Avalanche activity per danger level. The AAI considers all types of avalanches independent of snow conditions and trigger type; median value per day is given. Moreover, the number of days with either natural or human–triggered avalanches, at least size 2 or larger is shown.**

| Danger level | Number of days | Number of days with AAI > 0 (proportion in %) | AAI Median | Number of days with natural avalanches (≥ size 2) (proportion in %) | Number of days with human–triggered avalanches (≥ size 2) (proportion in %) |
|---|---|---|---|---|---|
| 1–Low | 303 | 26 (8.6 %) | 0.15 | 19 (6.3 %) | 7 (2.7 %) |
| 2–Moderate | 1766 | 437 (25 %) | 0.21 | 286 (16 %) | 144 (8.2 %) |
| 3–Considerable | 1366 | 741 (54 %) | 1.0 | 479 (35 %) | 341 (25 %) |
| 4–High | 94 | 93 (99 %) | 21 | 89 (95 %) | 36 (38 %) |
| 5–Very High | 4 | 4 (100 %) | 88 | 4 (100 %) | 0 (0 %) |

Whereas the number of natural avalanches steadily increased with increasing danger level, the relation differed for the human-triggered avalanches – mainly at the higher danger levels. The proportion of days with at least one human-triggered avalanche (≥ size 2) prominently increased from *1–Low* to *3–Considerable* (Table 2), about tripling from one danger level to the next; it less prominently increased to *4–High*, but was 0 at *5–Very High* – indicating that fewer people expose themselves to the hazard. The respective proportions were about 3 %, 8 %, 25 %, 38 % and 0 %.

Hence, triggering at the danger levels *1–Low* and *2–Moderate* was rather rare. In case a human–triggered avalanche was observed (≥ size 2) when the forecast avalanche danger level was *1–Low* or *2–Moderate*, this avalanche was in most cases (67 % and 68 %, respectively) the only one. In the case of natural avalanches, these proportions were lower, 47 % for *1–Low* and 48 % for *2–Moderate*.

The proportion of days with natural avalanches was higher than the proportion of days with human-triggered avalanches at all danger levels – even the lower ones (Table 2). The higher proportions of days with natural avalanches are primarily related to the occurrence of natural wet-snow avalanches at the lower danger levels. Below we will consider avalanche activity with regard to snow conditions and type of triggering in more detail.

## 3.2 Avalanche size

Smaller avalanches were more common than large avalanches – irrespective of the danger level (Table 3, Figure 3a). The majority of the avalanches recorded (9649 out of 13,745) were size 2 avalanches. This size was the most frequent at all danger levels. Size 3 and size 4 avalanches were less frequent at all danger levels. In other words, for sizes 2 to 4, the frequency of occurrence decreased with increasing avalanche size. The overall frequencies were 4 %, 70 %, 23 % and 2 % for the sizes 1 to 4, respectively (Table 3, bottom row).

**Table 3: Frequency of avalanche size per danger level (proportion in %).**

| Danger level | | Number of avalanches | | | | |
| | | Avalanche size class | | | | |
| | Number of days | 1 | 2 | 3 | 4 | Total |
|---|---|---|---|---|---|---|
| 1–Low | 303 | 3 (5.8 %) | 40 (77 %) | 8 (15 %) | 1 (1.9 %) | 52 |
| 2–Moderate | 1766 | 73 (5.5 %) | 977 (73 %) | 262 (20 %) | 18 (1.4 %) | 1330 |
| 3–Considerable | 1366 | 299 (4.8 %) | 4608 (74 %) | 1277 (20 %) | 82 (1.3 %) | 6266 |
| 4–High | 94 | 220 (3.7 %) | 3956 (67 %) | 1590 (27 %) | 166 (2.8 %) | 5932 |
| 5–Very High | 4 | 5 (3.0 %) | 68 (41 %) | 60 (36 %) | 32 (19 %) | 165 |
| Total | 3533 | 600 (4.4 %) | 9649 (70%) | 3197 (23 %) | 299 (2.2 %) | 13,745 |

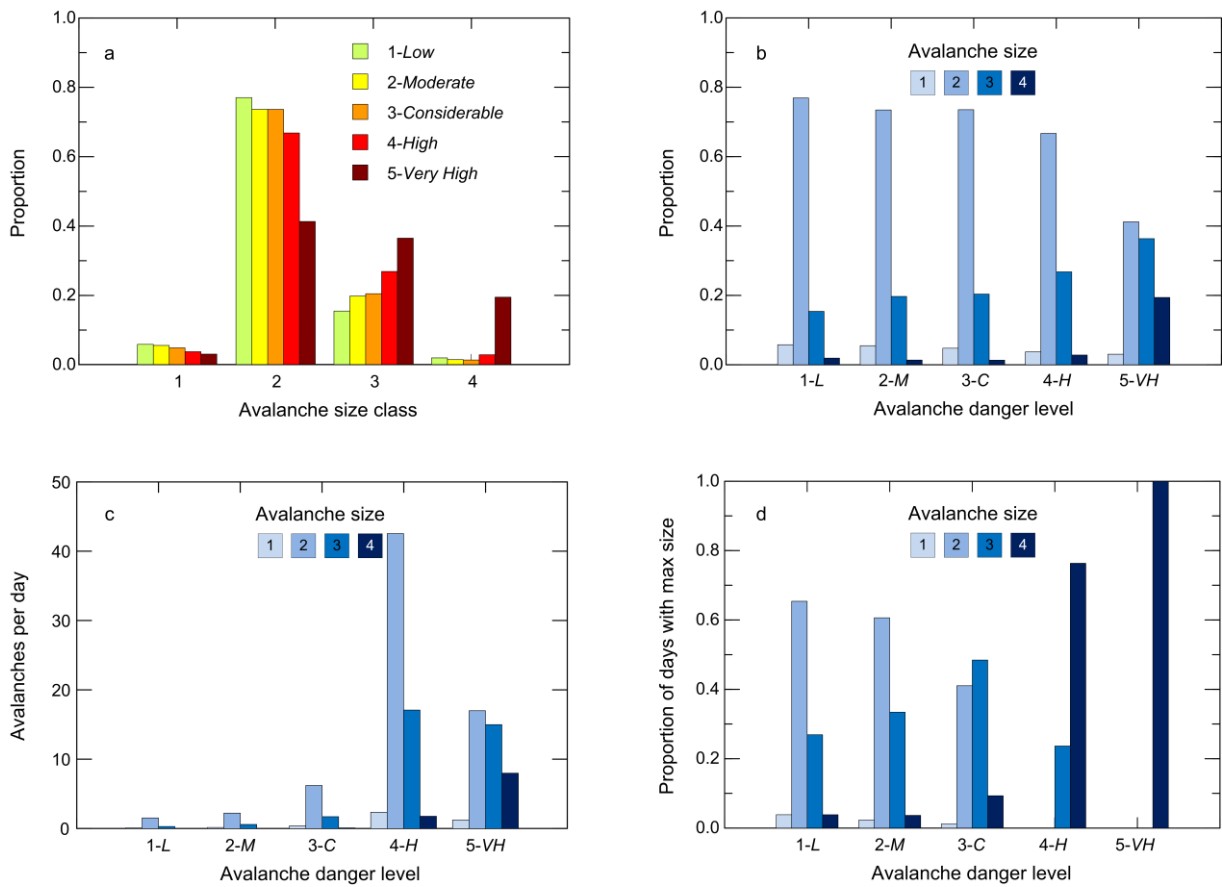

Figure 3: Avalanche size per danger level. (a) Relative frequency of avalanche sizes (1 to 4) at the danger levels *1–Low* to *5–Very High*. (b) Distribution of avalanche size for each danger level (same data as in Fig. 3a). (c) Average number of avalanches per day by size class for each danger level, for days with AAI > 0. (d) Frequency of days when the largest observed avalanche was either size 1, 2, 3 or 4 for each danger level. *N* = 13,745 avalanches on 1301 days.

The distributions of avalanche sizes per danger level (Figure 3b) did not significantly differ for the danger levels 1–*Low* to 3–*Considerable* (see Supplement, Table B). For 4–*High* the distribution looked similar, still the size 2 avalanches were the most frequent followed by size 3 avalanches, but the distribution was statistically different (*U*-test, $p < 0.001$). At danger level 5–*Very High* the distribution was clearly different with relatively more size 4 avalanches and less size 2 avalanches.

Overall, about 80-90 % of the avalanches were size 2 or 3, whereas size 1 and size 4 avalanches were always rare (≲5 %), except at *5–Very High* when about 19 % of the recorded avalanches were size 4 avalanches. Overall, size 2 avalanches tended to decrease, while size 3 and 4 avalanches tended to increase with increasing danger level (Figure 3b).

The proportion of size 2 avalanches was similar (73 to 77 %) at danger levels *1–Low* to *3–Considerable* (proportion test, $p > 0.57$), but clearly decreased at danger levels *4–High* (67 %; $p < 0.001$) and *5–Very High* (41 %; $p < 0.001$). On the other hand, the proportion of size 3 avalanches increased from 15 % at *1–Low* to 20 % at both *2–Moderate* and *3–Considerable* to eventually 27 % at *4–High* (Table 3). The latter increase was statistically significant (proportion test, $p < 0.001$). Size 4 avalanches were relatively most frequent at danger level *5–Very High,* about 5 times more frequent than at *4–High* and about 15 times more frequent than at *3–Considerable* (Figure 3c); the increase was statistically significant (proportion test, $p < 0.001$) .

At danger levels *1–Low* and *2–Moderate* avalanches were generally not smaller, but were simply less frequently observed. There was a substantial increase in avalanche occurrence with increasing danger level (Figure 3c) as also reflected in the strong increase of the avalanche activity index (Figure 2). The average total number of avalanches per day was 2, 3, 8, 64 and 41 for the danger levels *1–Low* to *5–Very High*, respectively. Hence, in general, the frequency rather than the size increased with increasing danger level. The median avalanche length was 164, 154, 163 and 198 m for the danger levels *1–Low* to *4–High*, respectively (see Supplement, Figure B). Hence only at danger level *4–High* the avalanches were about 25 % longer, the difference was statistically significant (*U*-test, $p < 0.001$). Avalanche length was weakly correlated with the danger levels (*1–Low* to *4–High*) ($r_s = 0.13$).

Whereas size 2 avalanches were clearly the most frequent at the danger levels *1–Low* to *4–High* and the size distributions looked partly similar, the largest avalanche observed at a given day increased with increasing danger level (Figure 3d). At the danger levels 1–*Low* and 2–*Moderate* the largest avalanche observed was on most days a size 2 avalanche, whereas at the danger levels 4–*High* and 5–*Very High* the largest avalanches were mostly or even always size 4 avalanches, respectively.

### 3.3 Snow conditions

For about half of all avalanches (56 %)  snow conditions were reported as dry, for 30 % as wet and for 6.3 % as mixed; for the remaining 8 % (1072 avalanches) the type of avalanche snow was unknown, i.e. not recorded. As the distributions of avalanche sizes were similar for the mixed and unknown conditions (*U*-test, $p = 0.822$), we merged these two groups. For the three groups (dry, wet, mixed/unknown), again about 70 % were size 2 avalanches and about 20 % size 3 avalanches (Figure 4a). The distributions of avalanche sizes were similar and comparable to the overall distribution (Table 3), but found to be statistically different (*U*-test, $p < 0.001$). Differences in size distribution with regard to snow conditions were however small. For size 2 avalanches, the proportions were even the same (71 %) for both dry and wet-snow conditions (proportion test, $p = 0.54$). The proportion of wet-snow avalanches of size 3 was slightly larger (25 %) than the corresponding proportion of dry-snow avalanches (22 %; $p < 0.001$). Most size 4 avalanches were recorded for mixed or unknown conditions, relatively twice as many as for dry-snow or wet-snow conditions. In general, for mixed or unknown conditions, the avalanche size seems to be somewhat larger. Relatively fewer size 2 and more size 3 and 4 avalanches were reported. In fact, the median avalanche length was 167,

and 190 m for dry-snow, wet-snow and mixed/unknown conditions, respectively. The avalanche length distributions were similar for wet-snow and unknown/mixed conditions (*U*-test, $p = 0.55$), but both different from the dry-snow conditions (*U*-test, $p < 0.001$). So far, in Figure 4a, we have considered all avalanches irrespective of the type of triggering. In the following, we will only consider natural avalanches.

Whereas overall the size distributions are not very different (Figure 4a), some differences emerge when considering the size distribution per danger level for dry-snow and wet-snow conditions (Figure 4b,c). For instance, at *1–Low* on a day with wet-snow avalanches also size 3 and size 4 avalanches were recorded. Avalanches tended to be larger and were also more frequent (Figure 4d). On the 10 days with natural wet-snow avalanches, the number of avalanches was 18 and the total AAI was 15.2. Whereas on the 6 days with natural dry-snow avalanches, the number of avalanches was 8 and the total AAI was 0.71. Hence, the AAI was on average more than ten times larger for wet-snow than for dry-snow conditions. On the other hand, as shown in Figure 4d the average number of avalanches per day was 1.3 for dry-snow and 1.8 for wet-snow conditions; this difference is not particularly large. In fact, the difference in the number of avalanches per day is statistically not significant (*U*-test, $p = 0.39$).

At the danger levels *1–Low* to *3–Considerable* there were relatively more size 3 and size 4 wet-snow avalanches recorded than dry-snow avalanches. Whereas for dry-snow conditions the proportion of size 2 avalanches decreased and the proportion of size 3 avalanches increased with increasing danger level, this tendency was not evident for wet-snow conditions.

Natural avalanches under wet-snow conditions tended not only to be larger, but there were also more wet-snow avalanches than dry-snow avalanches observed on a given day. The average number of natural avalanches per day with a given danger level was larger for wet-snow than for dry-snow conditions (Figure 4d). For wet-snow avalanches the average numbers were: 1.8, 2.5, 7.7 and 56 for danger levels *1–Low* to *4–High*, for dry-snow conditions: 1.3, 2.2, 4.8 and 22. For the danger levels *3–Considerable* and *4–High*, the difference in the number of avalanches per day was statistically significant (*U*-test, $p = 0.001$). Hence, the average number of avalanches per day was about 1.6 and 2.6 times larger at *3–Considerable* and *4–High*, respectively, for a day with wet-snow avalanches compared to a day with dry-snow avalanches. Overall, the 3044 natural dry-snow avalanches were recorded on 482 days, the 3955 natural wet-snow avalanches on only 331 days; i.e. on average almost twice as many wet-snow than dry-snow avalanches per day were recorded: 12 vs. 6.3. The difference in the number of avalanches per day between dry- and wet-snow conditions was statistically significant (*U*-test, $p = 0.001$) Hence, under wet-snow conditions, avalanches were not only larger, but also more frequent compared to dry-snow conditions.

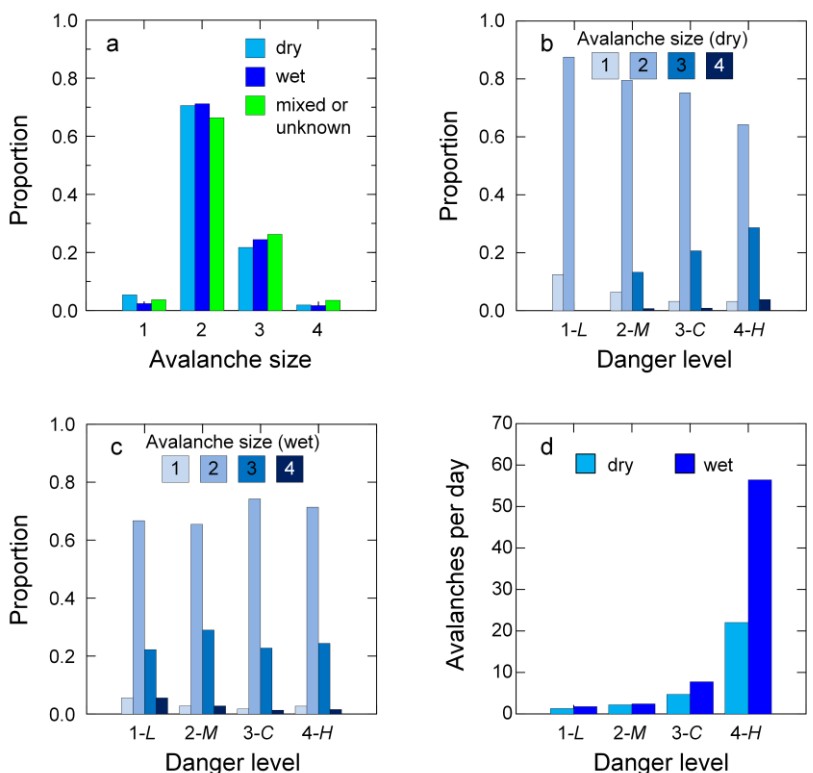

**Figure 4: (a) Avalanche size distribution for dry-snow avalanches (*N* = 7748), wet-snow avalanches (*N* = 4056) as well as for avalanches where the snow type was either recorded as mixed or it was not recorded at all (*N* = 1941). Avalanche size distribution per danger level (*1–Low* to *4–High*) for (b) natural dry-snow natural avalanches (*N* = 3044) and (c) natural wet-snow avalanches (*N* = 3955). (d) Number of avalanches on a day with corresponding avalanche activity per danger level for dry-snow and wet-snow conditions.**

## 3.4 Type of triggering

Whereas above we have compared avalanche activity with regard to snow conditions and primarily focused on natural releases, in the following we will only consider dry-snow avalanches and focus on the type of triggering: i.e. compare natural to human-triggered avalanches. For dry-snow conditions, there were about three times more natural (*N* = 3044) than human-triggered avalanches (*N* = 1036). For both natural and human-triggered avalanches, size 2 avalanches were most frequently observed, in 70 % and 74 %, respectively (Figure 5a), followed again by size 3 avalanches (24 % and 15 %). Hence, the frequency distribution was again similar, but statistically significantly different (*U*-test, $p < 0.001$). There were relatively more human-triggered avalanches of size 1 and 2, yet more natural avalanches of size 3 and 4. For instance, size 4 avalanches were 1.7 times more frequent among the natural than the human-triggered avalanches. Still, there were 17 human-triggered size 4 avalanches recorded. In summary, natural dry-snow avalanches tended to be larger than human-triggered dry-snow avalanches.

In total, 27 % of the dry-snow natural avalanches were size 3 and size 4 avalanches, whereas the corresponding proportion
was only 17 % among the human-triggered avalanches (proportion test, $p < 0.001$).

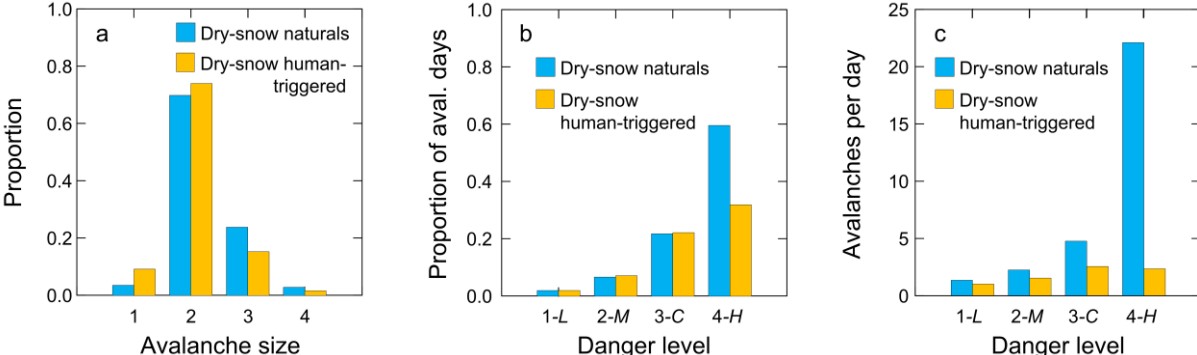

**Figure 5: Type of triggering for dry-snow avalanches. (a) Avalanche size distribution for natural and human-triggered avalanches. (b) Proportion of days with either natural or human-triggered avalanches with regard to danger level. (c) Number of avalanches per day vs. danger level.**

For the dry-snow human-triggered avalanches the size distributions at the four danger levels *1–Low* to *4–High* were similar to

the overall distribution shown in Figure 5a. For instance, at danger level *3–Considerable* and *4–High*, the proportion of size 3

avalanches was 0.74 and 0.73 (proportion test, $p = 0.77$). For size 4 avalanches the corresponding proportions were 0.14 and

0.19 (proportion test, $p = 0.36$). Moreover, the median avalanche length was 154, 139, 128 and 142 m at the danger levels *1–*

*Low* to *4–High.* The corresponding length distributions per danger level were statistically not different (pairwise *U*-test, p >

0.19). Hence, avalanche size did not increase with increasing danger level ($r_s = 0.004$).

We then considered the relative frequency of avalanche days per danger level (Figure 5b). Human-triggered as well as natural

dry-snow avalanches were rare when the danger was rated as *1–Low*. Only in 6 out of 303 days (2 %) with *1–Low* a human-

triggered avalanche was recorded, and on 6 days a natural avalanche (considering all size classes). In total, there were 6 human-

triggered and 8 natural avalanches on 11 individual days, i.e. typically there was one avalanche per day when there were dry-

snow avalanches at all at *1–Low*. Hence, avalanche occurrence at *1–Low* is unlikely, irrespective of the type of triggering. The

proportion of days with human triggered dry-snow avalanches increased to 7.2 %, 22 % and 32 % for days with forecast danger

level of *2–Moderate* to *4–High*, respectively. For natural dry-snow avalanches, the corresponding percentage values were 2 %,

6.7 %, 22 % and 60 %.

Not only the proportion of avalanche days, but also the average number of avalanches per day prominently increased with

increasing danger level (Figure 5c). For the human-triggered avalanches at *2–Moderate* the average number per day, when at

least one avalanche was recorded, was 1.5, at *3–Considerable* 2.5, but at *4–High* it slightly decreased to 2.3. For the natural

dry-snow avalanches, which are more closely related to the release probability and do not depend on the presence of people,

the increase was more prominent: 1.3, 2.2, 4.8, and 22 natural avalanches per day with danger rating *1–Low* to *4–High*, respectively. This corresponds to about a 2, 4 and 17 times increase from *1–Low* to the higher levels.

If not only dry-snow natural avalanches are considered, the increase of the average number of avalanches per day is even more prominent. Considering all natural avalanches irrespective of the snow conditions (i.e. dry, wet, mixed and unknown), the number of avalanches per day was 2.1, 2.5, 6.0, 48 and 33 for the danger levels *1–Low* to *5–Very High*, respectively. The higher number of natural avalanches per day, when considering all types of snow conditions instead of dry-snow only, reflects the finding that generally more wet-snow than dry-snow avalanches were observed at a given danger level as shown above in

Figure 4d.

## 4    Discussion

We analyzed a data set of visually observed avalanches from the region of Davos (Switzerland). Even though the data set was collected with the aim to record all (or at least as many as possible) avalanches in the region of Davos, also our data set certainly cannot provide the complete picture of avalanche activity. Small avalanches may be underreported in general. Moreover,

avalanche records based on visual observations are known to be biased since during times of poor visibility it is often difficult, and sometimes even impossible, to accurately outline the avalanche extent or record the release date (van Herwijnen and Schweizer, 2011). Only with remote avalanches detection systems the observation bias during storms can be overcome, at least with regard to the temporal resolution (Lacroix et al., 2012; Ulivieri et al., 2011). A good spatial resolution, can only be achieved with remote sensing from satellites (Eckerstorfer et al., 2017). However, presently the temporal resolution over the

Alps is still too poor for operational purposes, since only every 6 days images are acquired from the same orbit, which is necessary for change detection. Hence, when we provide the number of avalanches per day observed in our study region, this number should be considered as a lower limit, since with visual observation a full coverage of the study area cannot be achieved.

Moreover, there may be other biases as it is, for instance, easier to record wet-snow than dry-snow avalanches. Also, the level

of reporting varied during the 21 winter seasons. There were relatively more observations in the period of 2004-2005 to 2018-2019, than in the first 6 years of our study period. Still, size 2 and size 3 avalanches were the most frequent ones in both periods. The proportion of size 2 avalanches was about 70 % in both periods (proportion test, $p = 0.93$). The proportion of size 3 avalanches was 22 and 23 % during the first and second period, respectively, again not significantly different ($p = 0.28$). Hence, the key characteristics did not change much.

With 13,918 avalanches the data set is extensive and covers many different avalanche situations; small as well as large avalanches, single avalanches as well as records from intense storms with many avalanches – in short, it seems a rather complete data set. A much smaller data set with also mapped avalanche perimeters for the surroundings of the village Zuoz in the lower Engadine (Swiss Alps) was analyzed by Stoffel et al. (1998). In France, the "Enquête permanente sur les avalanches" (EPA)

is an extensive data set including the avalanche events in approximately 5,000 major paths in the French Alps and the Pyrenees
(e.g., Eckert et al., 2010). Primarily large avalanches that can threaten infrastructure are recorded. Hence this data set seems biased towards very large avalanches and less suited for our purpose. Other extensive data sets were record along mountain passes, e.g. along the Milford road in New Zealand (Hendrikx et al., 2005). They found a significant lack of smaller sizes in their size class distribution, which contained 1842 avalanches. Since only larger avalanches (size ≥ 2.5) are relevant for road safety, smaller avalanches were under-reported. As similar underestimation of size 1 and size 2 avalanches was present in
avalanche records observed along the highway crossing Rogers Pass (Canada) (McClung and Schaerer, 1981). Hence, our data set seems to be one of the most comprehensive ones – providing unique insight into avalanche activity. In general, the type of recording and the potential impact (e.g., whether the avalanche hit the road and/or caused damage) may represent the most relevant biases in avalanche data sets.

The avalanche activity as described based on our data set, also depends on terrain characteristics and snow climate. Hence, the
characteristics of avalanche activity in our study area cannot simply be transferred to other mountain regions. Nevertheless, we suppose the avalanche size distribution for a given danger level to be fairly robust and rather independent of terrain and climate characteristics. However, the frequency and intensity of avalanche occurrence will certainly vary between regions.

The proposed size classification based on perimeter data had previously been used to study indicator avalanches in the regions of Davos (Schweizer et al., 2012). With the suggested classification criteria, the resulting median length (Table 1) is well in
line with the typical values associated with the corresponding size classes provided by the European avalanche warning services (EAWS, 2019b). They indicate typical length categories of <50 m, 50-200 m, several 100 m, 1-2 km for size classes 1 to 4, respectively.

We then compared avalanche activity to the forecast danger level. Again, this is far from perfect as one would need the verified danger level to compare with. Whereas verification at the lower danger levels can be done with numerous snow instability
tests (Schweizer et al., 2003), measurements (Reuter et al., 2015) or by expert opinion (Techel and Schweizer, 2017), at the higher danger levels (*4–High* and *5–Very High*) avalanche activity is the most reliable hazard indicator.

As no consistent verification data existed for the entire data set, we had to use the forecast. Still, we tried to remove some obvious outliers such as days with forecast danger level *4–High* but no avalanche records. This quality check should not be considered as comprehensive verification. In total, we changed less than 4 % of the danger ratings and mostly increased the
danger level (Table A in the Supplement). In contrast, most verification studies showed a forecast accuracy of about 70-80 % and a trend to over-forecasting (e.g., Techel and Schweizer, 2017). Hence, our avalanche danger data are still biased, yet also reflect some past and recent practice of applying the danger levels. For example, the danger level *4–High* was relatively rarely forecast (on less than 3 % of the days). This may partly be explained by the location of Davos, which is somewhat protected from major storms, but also relates to forecast practice in Switzerland (Techel et al., 2018). However, it is also remarkable that
similar avalanche activity was often differently rated for dry-snow and wet-snow conditions – at all danger levels. This finding

adds to the list of inconsistencies in avalanche warning as recently reported in several studies (Clark and Haegeli, 2018; Lazar et al., 2016; Techel et al., 2018).

Correcting the forecast danger level based on avalanche activity, while analyzing the relation between danger level and avalanche activity, may seem questionable. The main effect of the correction procedure was that the median AAI for days with 4–*High* increased from about 10 to about 21. For days with 3–*Considerable*, the median AAI was 1 and did not change due to the corrections. Hence, the difference in avalanche activity between 3–*Considerable* and 4–*High* was already very prominent before the correction procedure (before and after the corrections: *U*-test, $p < 0.001$). With regard to avalanche size, the effects are less prominent. Size 2 avalanches were the most frequent ones at danger levels 1–*Low* to 4–*High* before and after the corrections.

The avalanche size distribution we found was remarkably robust with regard to different data stratifications. In particular, the size distribution did not change substantially with the danger level (Figure 3). In other words, for our data set, avalanche size did not prominently increase with increasing danger level, at least for the danger levels *1–Low* to *4–High*; size 2 avalanches were the most frequent (Figure 3b). This finding seems somewhat surprising, given that the avalanche danger level is characterized by a combination of the probability of avalanche occurrence and expected avalanche size (Meister, 1995); it suggests that avalanche size may be of secondary importance compared to snowpack stability and its distribution when assessing the danger level (Techel et al., 2020). Hence, it seems unlikely that the typical (or most frequent) avalanche size is decisive for choosing between four different danger levels for one given snow stability scenario as suggested recently by Müller et al. (2016a). Also, in the conceptual model of avalanche hazard (CMAH) a frequency-magnitude (or likelihood-size) matrix was suggested to estimate avalanche hazard (Statham et al., 2018).

On the other hand, considering the largest avalanche recorded on a given day as suggested by Techel et al. (2020) showed more prominent differences between the danger levels *1–Low* to *2–Moderate* compared to *4–High* to *5–Very High* (Figure 3d). Hence, the maximum expected avalanche size may be useful to differentiate between some hazard situations (Techel et al., 2020). Clark and Haegeli (2018) also reported maximum expected size to be the second most relevant factor for selecting a hazard rating.

Moreover, our findings on avalanche size are in line with the results of a study on avalanche incidents in relation to the danger rating. Harvey (2002) reported that length, width and fracture depth of human-triggered avalanches were very similar at the danger levels *1–Low* to *4–High*, hence did not increase with increasing danger level; the median length was 200 to 250 m, the width 50 to 60 m, which corresponds to avalanche size 2, in agreement with our analysis. However, when he considered all avalanches that had caused damage, i.e. not only human-triggered avalanches, but also avalanches that had destroyed trees or infrastructure, he found avalanche size to be larger at the danger levels *4–High* and *5–Very High* than at the lower danger levels. Logan and Greene (2018) recently also related avalanche size to danger level. They reported that size 2 avalanches

(destructive size) were the most frequent size at the danger levels *2–Moderate, 3–Considerable* and *4–High* – in agreement with our findings.

The relative frequency of the avalanche size classes 2, 3 and 4 were 70 %, 23 % and 2 %, respectively. Hence the frequency of avalanche sizes 2 to 4 decreased with increasing size (Table 3, bottom row). Stoffel et al. (1998) also reported decreasing frequency of occurrence for their avalanche size classes medium, large and very large. These findings are in line with the magnitude-frequency relation of most natural hazard events including earthquakes for which the relation is known as Gutenberg-Richter law (Jentsch et al., 2006). Several other studies have shown frequency-size power-laws for snow avalanches (e.g., Birkeland and Landry, 2002; Faillettaz et al., 2004). The fact that size 1 avalanches were not the most frequent, as theoretically should be the case, is probably related to an observation bias: small avalanches may often not be mapped, in particular when other larger avalanches occur, or cannot be mapped at all if they are too small. In fact, when considering avalanche length and plotting a cumulative curve of events, avalanches shorter than 100 m seem to be underrepresented. This type of observation bias is also present in earthquake catalogues (e.g., Woessner and Wiemer, 2005).

The average number of observed natural avalanches strongly increased with increasing danger level: 2.1, 2.5, 6 and 48 for the danger levels *1–Low* to *4–High*, respectively. This suggests that the differentiation between the lower two danger levels cannot be based on avalanche occurrence. Also, the relative increase from one danger level to the other is increasing – suggesting an exponential increase of the hazard. Previously, it was suggested that the hazard would double from one level to the other (Munter, 2003). Using accident data a 2- to 3-fold increase was shown (Pfeifer, 2009; Techel et al., 2015), whereas a survey among North American avalanche professionals suggested a 10-fold increase of triggering probability when the regional danger increases by one level (Jamieson et al., 2009b). Based on avalanche observations from Colorado, Elder and Armstrong (1987) assigned avalanche frequencies per day to the four danger levels that were in use at those times: 0-3, 4-9, 10-20 and ≥21.

The prominent, non-linear increase of avalanche occurrence with increasing danger level reflects that, according to its definition, the avalanche danger level increases with decreasing snow stability. With decreasing snow stability, the frequency of locations with a potential weakness where an avalanche may be released increases. However, the suggested increase of weaknesses in the snowpack can only be assessed with spatial variability studies (e.g., Reuter et al., 2016), which are most appropriate to determine the spatial distribution of instabilities. For example, Schweizer et al. (2003) reported an increase of the proportion of poorly rated profiles from virtually 0 % to 24 % to 53 % for the danger levels *1–Low* to *3–Considerable*, respectively. Their observations correspond to our finding that the number of natural dry-snow avalanches doubled from *2–Moderate* to *3–Considerable*, and increased almost three times for wet-snow avalanches. Whereas the number of natural dry-snow avalanches consistently increased with increasing danger level, this was not the case for the human-triggered avalanches. The frequency of human-triggered avalanches did not increase from *3–Considerable* to *4–High*. This finding does not mean that triggering becomes less likely but rather reflects terrain usage and the effect of avalanche warnings. In fact, Techel et al. (2015) showed a decrease in ski touring activity already from *2–Moderate* to *3–Considerable*, and even more prominently from *3–*

*Considerable* to *4–High.* On the other hand, Wäger and Zweifel (2008) found no decrease in frequency usage from *2–Moderate* to *3–Considerable* when considering off-piste skiing.

At danger level *4–High*, the average number of natural avalanches per day in our study region was about 20 for dry-snow avalanches and about 50 for wet-snow avalanches. If we assume that '*many*' natural avalanches are typically observed at the danger level *4–High* (EAWS, 2019a), we may conclude that 10-20 avalanches per 100 km$^2$ have to be expected, considering
some underreporting in our study area of about 300 km$^2$, where about two thirds are avalanche terrain. Hence, our analyses suggest that in the terrain typical of our study area at least about 10 natural avalanches have to be observed for verifying the danger level *4–High*. Based on satellite images, Bründl et al. (2019) recently analyzed the avalanche activity during a major avalanche cycle in January 2018 when the danger level was *5–Very High*. They analyzed the number of size 4 avalanches per area; for an area of 250 km$^2$ the number of size 4 avalanches ranged from less than 29 to up to 202 – roughly consistent with
our observations.

With regard to the definition of the avalanche danger by snow stability, its spatial distribution and avalanche size, we would like to point out that the spatial distribution does only refer to the frequency of locations with very poor snow stability where avalanches can initiate from. In other words, where the points with very poor snow stability are located in space is irrelevant, only their frequency counts, when deciding on a given danger level. Moreover, there is a difference between the probability of
avalanche release, which is a local property related to local snow instability as recently re-visited by Reuter and Schweizer (2018), and the avalanche occurrence probability, which depends on scale and is the result of stability and its distribution (frequency of triggering spots) for a given area. Moreover, there is a third probability, not to be confused with the two previously mentioned, namely the probability of triggering an avalanche faced by an individual who travels in avalanche terrain on a given day with a given danger level, which also depends on scale (terrain travelled) and terrain choices.

Finally, when the definition for *2–Moderate* danger states that "*Large natural avalanches are unlikely*" (EAWS, 2019a), this definition could as well be modified to "*Natural avalanches are unlikely*" since the probability for any size of natural avalanche at *2–Moderate* is less than 5 %, which according to the IPCC (IPCC, 2014) is best described by '*very unlikely*'. Likewise, the formulation in the definition of *1–Low* "*Only small and medium avalanches are possible.*" is not appropriate and should be modified. Hence, our analyses of the avalanche activity may be used to improve the descriptions in the avalanche danger scale.

**5    Conclusions**

We quantified some of the key characteristics such as frequency and size of avalanches at a given danger level. To this end, we analyzed a unique data set of 21 years of visually observed avalanche records from the surroundings of Davos (Eastern Swiss Alps, 300 km$^2$), consisting of the mapped outlines of 13,745 avalanches, and compared the characteristics of avalanche activity to the regional danger level as forecast on 3533 days.

The proportion of days with natural avalanches at a given danger level substantially increased with increasing danger level. Also, the overall number of avalanches per day prominently increased, which reflects that with increasing danger level snow stability decreases and the frequency of locations with a potentially critical weakness in the snowpack increases. The recorded number of avalanches per day in our study region (300 km$^2$) was 2, 3, 8 and 64 for the danger levels *1–Low* to *4–High*, respectively.

The relative frequency of the four avalanche size classes did not substantially change with increasing danger level, neither for human-triggered nor for natural avalanches, except for danger level *5–Very High*. In other words, avalanche size did not increase with increasing danger level: the most frequent avalanches were size 2 avalanches at any danger level. This suggests that avalanche size may be of secondary importance compared to snowpack stability and its distribution when assessing the danger level. Only in certain situations avalanche size may be decisive – and rather by considering the maximum expected size. Still, the absolute number of very large avalanches (size 4) per day prominently increased, namely by 20 times from *3–Considerable* to *4–High*.

At a given danger level the frequency of natural avalanches was typically larger for wet-snow conditions than for dry-snow conditions and wet-snow avalanches tended to be larger – potentially reflecting inconsistent usage of the danger scale in Switzerland. Based on our findings, we propose revisiting the definitions of the danger scale and possibly quantifying some of the descriptions. For example, we suggest that '*many avalanches*' may mean on the order of at least about 10 avalanches per 100 km$^2$.

We are aware that visual observations are notoriously incomplete. Hence, our results should be challenged by similar analyzes with similarly extensive data sets. In future, more comprehensive data sets based on remotely-sensed data and results from avalanche detection systems may allow better founded analyses.

Finally, our analyses confirm that different avalanche situations are typically condensed into one specific danger level, which results in a loss of information. Hence, avalanche warning services are encouraged to describe the danger as best as they can, and not only provide the danger level. Likewise, risk assessment in avalanche education should not focus solely on the danger level, or the release probability, but as well include the potential consequences.

**Data availability**

The data sets of avalanche observations and avalanche danger will become available at www.envidat.ch.

**Author contributions**

JS, CM and BR designed the study. AS, FT, JS and CM extracted and curated the data. JS analyzed the data and prepared the manuscript with contributions from all co-authors.

**Competing interests**

The authors declare that they have no competing interests.

**Acknowledgements**

We would like to thank everyone who contributed to the avalanche occurrence data set, in particular the interns of the avalanche warning service who mapped most of the avalanches. We are grateful for the constructive comments by the two reviewers and the editor that helped to improve the paper.

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
