# Peer review of "On the relation between avalanche occurrence and avalanche danger level"

_The Cryosphere, 2019_

## Referee Comment (RC1) · Rune Engeset (Referee) · 3 Oct 2019

Review of the manuscript *Schweizer, J., Mitterer, C., Techel, F., Stoffel, A., and Reuter, B.: On the relation between avalanche occurrence and avalanche danger level, The Cryosphere Discuss., https://doi.org/10.5194/tc-2019-218*

The paper is of high scientific quality. It brings forward important quantitative results on the occurrence of snow avalanches in the Eastern Alps, and provides an important analysis and discussion of the implications for assessment and forecasting of avalanche danger. The study uses a unique and rich data set on observed avalanches and regional avalanche danger estimates, in order to reduce a critical knowledge gap that has limited the development of objective procedures for determining the avalanche danger level. The publication of the paper will contribute to improved workflows, standards and eventually better avalanche forecasting products in the future.

My recommendation to the editors is to publish the paper, after addressing the points below (minor revisions).

The language, figures and tables are generally of high quality and easy to read. The structure is easy to follow, and the balance between data, results and discussion is well suited for a publication. However, I recommend improving readability by splitting many long complex sentences into shorter sentences.

One aspect I was missing was an analysis and/or discussion of the general transferability of the results to other parts of the world, especially where terrain or climate conditions differ from the Davos region. One could assume that not only the total size of the study area matters, but rather the size of avalanche terrain area. Some terrain is not able to produce larger avalanches, while other types of terrain produce many large avalanches. Some climates produce many large natural avalanches, while others less so. If the authors could add these aspects to the discussion, it may be easier for the readers or future studies to generalise, add to, or test the results.

Now follows specific comments, with reference to line numbers in the manuscript:

**7 Add size of study area, number of avalanche observations and number of regional danger assessments.**

**13 Could the sentence ending in "…given day" be improved by adding "an danger level" at the end?**

**15 Could the sentence be improved by replacing "may allow revisiting" by "suggest reworking of"?**

**18 Add "according to our data" after "km$^2$"**

**24 Improve the sentence starting with "For these…"**

**35 Improve flow (order of words) of sentence**

**35-38: May also use the EAWS description "Avalanche danger is a function of snowpack stability, its spatial distribution and avalanche size" (https://www.avalanches.org/wp-content/uploads/2019/07/general-assembly-oslo_minutes_EAWS.pdf)**

**50-51 This description should be updated. The latest EAWS matrix specifically accounts for size (https://www.avalanches.org/standards/eaws-matrix/)**

**75 Explain if the size of the area of avalanche observations is equal to the size of the forecasting region**

**75 Add a short description of how these observations were obtained. Information is provided in the discussion chapter, but it would be logical for the reader to learn about the data upfront. Did the observations cover the entire 360 km$^2$? The entire winter season? All seasons with the same rigorousness?**

**107 Could replace "In other words, on" by "On average,"**

**109 Add a sentence at the beginning of the paragraph, about why the work described in the paragraph was carried out. E.g., "The forecasting data were scrutinized, in order to adjust danger levels to the most realistic values."**

**126 Replacing "Moreover, there were also days, 17 in total" with "This was also the case for 17 days" could improve the readability**

**165 Add a descriptor for the values, probably "median values"**

**175 Replace "for" by "to"**

**277 Replace "to" by "of"**

**289 Since the Eckerstorfer el al. study, the number of Sentinel-1 satellites has doubled with 1B in orbit. Thus, the statement of too poor temporal resolution is less valid today. I suggest to add this information to the sentence.**

**305 Spell out what is meant by "the potential impact"**

**374 Terrain usage probably also decrease from level 2 to 3, as well as from 3 to 4. It would be useful to add references, if these exist, on the differences in terrain usage between danger levels 1, 2, 3 and 4.**

**377 Explain in more detail how you arrive at the number 10.**

**384-389 I find parts of this paragraph unfinished. I would recommend arguing or substantiating why you make the statements "need to" and "should not". I would also recommend to put the sentence "The actual locations…" into context (e.g. the wordings of the NA/CMAH and EADS wrt. spatial distribution).**

*Rune Engeset, 3 October 2019.*

---

## Short Comment (SC1) · 3 Oct 2019

Lines 375 to 381 - In my humble opinion, these are the most important data to eventually discover: proportion of potential avalanche paths releasing the given avalanche problem for the forecasted spatial and temporal scale hazard rating.

These data will be very difficult, or impossible for size 1 avalanches, to acquire but are the most important for avalanche risk management practices, i.e. how many / much of the avalanche paths / terrain are expected to release the forecasted avalanche problem type today?

A very important paper and adds much to the field.

---

## Referee Comment (RC2) · Pascal Haegeli (Referee) · 23 Oct 2019

TC-2019-2018

**On the relation between avalanche occurrence and avalanche danger level**

Authors: J. Schweizer, C. Mitterer, F. Techel, A. Stoffel and B. Reuter

Review by Pascal Haegeli, Simon Fraser University, Vancouver BC, Canada
Oct. 23, 2019

The present manuscript describes an analysis that takes advantage of a large existing dataset of observed and mapped avalanches to explore the relationship between avalanche danger levels and avalanche occurrence in the Davos region of Switzerland. The research is of high quality, and the results contribute valuable insights to the current discussion on avalanche forecasting practices and consistency. The various analyses described in the manuscript offers useful information on the role avalanche size in avalanche forecasting, and the recommendation on the number of expected avalanches at danger rating level High has the potential to be the starting point for making avalanche forecasting more objective by replacing the existing qualitative descriptors in the danger scale with more objective quantitative measures.

The manuscript fits well with the mandate of The Crysophere journal, and it will offer great value for avalanche safety researchers and practitioners. However, despite the obvious strength of the research, I believe that the manuscript has a few weaknesses that should be addressed before the manuscript is published. My concerns mainly relate to the presentation of the dataset correction procedures and the qualitative description of the results. I hope that my comment below are useful for making the manuscript even more impactful.

**GENERAL COMMENTS AND SUGGESTIONS**

**Correction procedure (Lines 109-140)**

Given that the objective of your paper is to examine the relationship between avalanche danger ratings and avalanche activity, manually changing danger ratings based on observed avalanche activity prior to analysis seems risky. While I do not necessarily disagree with the approach, I have the following recommendations for making it more transparent for the reader:

- To put the number of days with corrected the danger ratings into perspective, it would be useful to provide readers with counts and proportions of danger ratings prior to correction right at the beginning of this section. This information is currently only available for the corrected danger ratings (Table 2). Having this information upfront would help readers to understand how much of the dataset was modified.

- The description of the correction procedure refers several times to the fact that avalanche activity was 'unusually high' or 'unusually low'. However, you do never explicitly specify what your expectations regarding avalanche activity actually are and how you determined your thresholds (one exception is the recoding of moderate days with AAI > 1.0). Being more explicit about your criteria would make your procedure more transparent.

- I personally found the description of the correction procedure somewhat difficult to follow due to many details described in the text. I wonder whether a diagram (e.g., flow chart) showing which danger ratings were changed to what and for what reason would help the reader to better understand the magnitude of your changes and their potential impact on the subsequent analysis.

- The numbers in your description of the changes applied to danger with High and Very High danger ratings (Lines 111-116) do not seem to add up properly.

- Overall, you changed the danger rating in 122 of 3533 days, which only amounts to 3.5%. This seems like a rather small amount and my initially thought was that the correction procedure was unnecessarily complicated given that it will likely only have a minor impact on the analysis. However, it represents 9% of the days with avalanche activity, and, if I understand your descriptions correct, the number of days with danger ratings High and Very High changed substantially through the correction procedure. The number of days with a High danger rating was first reduced from 44 to 26 (-18) (Line 116) and then increased again to 94 (+68) (Line 141). This means that only 28% of the days with High danger ratings in the analysis dataset were originally assigned a High danger rating. Given that the High danger rating sample plays an important role in the subsequent analysis, I believe that the impact of the correction procedure on the nature of the dataset should be described more clearly.

- A brief discussion of the potential effect of the correction procedure on the analysis results in the discussion section would further acknowledge its impact. I think it is important to explicitly mention that there is potential for a bit of a circular argument here: You corrected the danger rating levels based on avalanche activity expectations to later analyze exactly this relationship.

**Description of analysis methods**

The section titled 'Data and methods' only includes descriptions of the derivation of the avalanche size, the danger rating dataset and the quality control and correction procedure but seems to completely skip a description of the actual analysis approach and statistical methods employed. This seems rather unusual. I believe the manuscript would benefit a short overview of the analysis approach that describes the measures used (e.g., avalanche activity index, proportion of days with avalanches, etc.) and how they relate to the components of avalanche hazard (e.g., snow stability, frequency of locations) in the methods section.

**Statistical support for qualitative descriptions**

Much of the description of the observed patterns are rather qualitative with some statistical tests here and there. I am wondering whether some of the statement could be supported with statistical test statistics. I believe that this would considerable strengthen the power of the manuscript.

**Description of study area**

On line 377, you provide recommendations about the number of expected avalanches at danger rating level 4-High (at least 10 per 100 km$^2$), and on line 416, you suggest that the term "many avalanches" should mean on the order of at least about 10 avalanche per 100 km$^2$. I believe that this is an interesting result. However, while you highlight that avalanche occurrence probability depends on scale as it is a combination of stability, its distribution within the forecast area and the size of the forecast area, it seems to me that the nature of the terrain in the forecast region would also have a substantial impact on the suggested number. I therefore wonder whether a more detailed description of the nature of the avalanche terrain in the study area (e.g., number of avalanche paths of different size, total extent of avalanche terrain) would offer valuable context for understanding the results and recommendations.

**Insight into avalanche warning practices**

In several places in this manuscript, you comment on the somewhat unexpected differences in the observed numbers of dry and wet avalanches at the same danger rating level (e.g., Lines 323-324, Lines 375-376). However, there is no explicit statement in the discussion or conclusion section that points out that these observations might indicate inconsistencies in forecasting practices. I think that a statement like this would fit nicely with the recent literature on avalanche forecasting inconsistency and further contribute to this research.

**LINE-SPECIFIC COMMENTS**

1. **Line 95 – Figure 1**
   The exponential increase presented in Figure 1 seems to be the direct consequence of the classification criteria presented in Table 1. I wonder whether plotting the log of avalanche area versus avalanche size class would be more useful to highlight that the approach classifies avalanche in the spirit of the Canadian size classification.
   The number of avalanches per class shown in the chart do not add up to the total number of avalanches given in the caption.

2. **Line 100**
   It might be useful to explicitly state that the weight of 0.81 is appropriate because it is highly likely that the avalanches without known triggers were likely natural avalanches.

3. **Line 191 – Table 3**
   It might be useful to add row percentages to the columns to better highlight the relationship between avalanche size distribution and danger rating.

4. **Line 192**
   The Kruskal-Wallis test only indicates whether there are any differences in the avalanche size distributions among all danger rating levels. You could follow-up with pairwise Wilcoxon rank-sum tests between adjacent danger rating levels to determine where exactly the differences are.

5. **Line 196 – Figure 3**
   I think it would be best if the proportion scales in all charts would range from 0 to 1 and be styled the same.

6. **Line 196 – Figure 3c**
   If I understand your analysis correctly, Fig. 3c depicts average or median number of avalanches of different sizes per day at different avalanche danger levels. A bar chart does not seem an appropriate way to display this information as bar charts are typically used to depict proportions. Populating the same layout grid with a series of box plots instead of vertical bars would represent not only the magnitude difference in number of avalanches between danger ratings and avalanche size, but also the range within the observations.

7. **Line 209**
   Can the statement "The median as well as the 90-percentile avalanche length did not increase for the danger levels 1-Low to 4-High." be substantiated with a statistical test result?

8. **Line 217 and Figure 4a**
   It seems odd to combine avalanches with unknown snow conditions with the mixed category in the snow conditions analysis (Section 3.3) as these are very different categories. I think it would be better to leave these avalanches out of the analysis all together.

9. **Line 227 and Figure 4d**
   The difference in the number of avalanches per day between dry and wet avalanches under avalanche danger level 1-Low shown in Fig. 4d seems minute. Can this statement be supported with a statistical test?

10. **Line 233**
    I assume that this discussion should be referring to the AVERAGE or MEDIAN number of avalanches per day.

11. **Line 241 – Figure 4d**
    Same comment as for Figure 3c

12. **Line 253**
    The statement "… size 4 avalanches were five times more frequent among the natural than the human-triggered avalanche." does not seem to be supported by Figure 5a.

13. **Line 256 – Figure 5c**
    Same comment as for Figure 3c and Figure 4d

14. **Lines 268-273**
    Same comment as earlier regarding the average/median number of avalanches per day.

15. **Line 293**
    I think it would be useful for the reader if the fact that no temporal trends in the avalanche size distribution were detected in the analysis dataset was included and substantiated in the initial description of the dataset.

16. **Lines 316-324**

   It seems to me that some of the explanations of the data correction procedure described in this paragraph should be included in the methods section.

17. **Lines 340-346**

   In this section, you refer to the avalanche size distribution of human triggered avalanches, and the topic comes up again on Line 407. However, I did not find this explicit analysis in your manuscript. If I read your manuscript correctly, you only analyzed the number of human triggered avalanches under different danger ratings but not their size distributions. It seems to me that such an analysis would nicely complement your existing analyses .

18. **Lines 355**

   Same comment as earlier regarding the average/median number of avalanches per day.

19. **Lines 378-381**

   The description of the results of Bründl et al (2019) is a bit confusing to me. What are the five frequency classes and how do they relate to the results presented in this paper?

20. **Line 389**

   Add "and terrain choices." at the end of this sentence.

---

## Short Comment (SC2) · 24 Oct 2019

Dear authors, this is a very interesting paper that I am looking forward to getting published. I would like to ask you, however, to explain in more detail how you derived your avalanche size classification based on the Canadian system:

You state that you used the Canadian classification system and I assume that you have used the typical path lengths for your classifications. If that is true, where are you thresholds? A size 2 avalanche for example has a typical path length of 100 m, while a size 3 has a path length of 1000 m. Where is the threshold path length between these two sizes?

I am also wondering in general if typical path length is a good measure for avalanche

size. You can have an avalanche that ran over a very long distance but basically did not entrain any snow. so the debris is fairly small then and so is likely its destructive potential.

Buehler et al (in discussion in TC right now) for example used debris area (width x length of the debris) as a measure for avalanche size. You show the median area of your avalanches per size class in Table 1. For size 1 for example, you get 544 m2, which is above the threshold of 500 m2 used by Buehler et al., as well as by Bruendl et al in their great report about the large avalanche cycle in Switzerland.

Thank you for considering my questions!

all the best, Markus Eckerstorfer

---

## Author Comment (AC1) · 30 Oct 2019

**Reply to comment by Markus Eckerstorfer on Avalanche Size Classification**

Dear Markus,

Many thanks for your comment on avalanche size classification. As you mention, we refer to the Canadian Avalanche Size Classification (McClung and Schaerer, 2006), which does not include avalanche width and length, but refers to destructive potential. However, length and width are the measures most people refer to – rather than area.

Hence there were a number of attempts to indicate avalanche length (and width) for the different size classes. This is most straightforward for size 2 avalanches since those are commonly seen as the typical human-triggered avalanches. For this type of avalanche, length and width are often reported. For example, Schweizer and Lütschg (2001) reported the median length and width of human-triggered avalanches, as 150 m and 50 m, respectively. Fatal avalanche were larger with a median length of 310 m and a median width of 80 m.

Often the typical length of size classes 1 to 3 are described with 10 m, 100 m, 1000 m (Stethem et al., 2003). However, this numbers cannot easily be used. For example, in case of size 3 avalanches 1000 m is obviously rather the upper limit, whereas 10 m for size 1 avalanches is rather the lower limit. It seems clear that for size 1 and 2 avalanches the typical length is rather several ten or a few hundred meters – and several hundred meters for size 3 avalanches. Numbers in these ranges are, for instance, given on the website of the European Avalanche Warning Services (EAWS, 2019) or in Schweizer et al. (2015).

We have adapted these lengths values and added width as an additional criterion. The threshold values we use are given in Table 1 of our manuscript.

As for avalanche length, several different thresholds were used for mapping avalanche area into avalanche size. One version, as you mention, is shown in Bühler et al. (2019).

For our classification based on length and width, we have in addition provided the corresponding avalanche areas, which might be useful for future classification attempts based on area.

Jürg Schweizer.

**References**

Bühler, Y., Hafner, E. D., Zweifel, B., Zesiger, M., and Heisig, H.: Where are the avalanches? Rapid mapping of a large snow avalanche period with optical satellites, Cryosphere Discuss., 2019, 1-21, https://doi.org/10.5194/tc-2019-119, 2019.

EAWS: Avalanche Size: https://www.avalanches.org/standards/avalanche-size/, access: 27 July 2019, 2019.

McClung, D. M., and Schaerer, P.: The Avalanche Handbook, 3rd ed., The Mountaineers Books, Seattle WA, U.S.A., 342 pp., 2006.

Schweizer, J., and Lütschg, M.: Characteristics of human-triggered avalanches, Cold Reg. Sci. Technol., 33, 147-162, https://doi.org/10.1016/S0165-232X(01)00037-4, 2001.

Schweizer, J., Bartelt, P., and van Herwijnen, A.: Snow avalanches, in: Snow and Ice-Related Hazards, Risks and Disasters, edited by: Haeberli, W., and Whiteman, C., Hazards and Disaster Series, Elsevier, Amsterdam, Netherlands, 395-436, 2015.

Stethem, C., Jamieson, J. B., Schaerer, P., Liverman, D., Germain, D., and Walker, S.: Snow avalanche hazard in Canada - a review, Nat. Hazards, 28, 487-515, https://doi.org/10.1023/A:1022998512227, 2003.

---

## Author Comment (AC2) · 15 Jan 2020

**Reply to Reviewer #1**

We thank the reviewer for the positive feedback and his constructive suggestions.

*The paper is of high scientific quality. It brings forward important quantitative results on the occurrence of snow avalanches in the Eastern Alps, and provides an important analysis and discussion of the implications for assessment and forecasting of avalanche danger. The study uses a unique and rich data set on observed avalanches and regional avalanche danger estimates, in order to reduce a critical knowledge gap that has limited the development of objective procedures for determining the avalanche danger level. The publication of the paper will contribute to improved workflows, standards and eventually better avalanche forecasting products in the future.*
*My recommendation to the editors is to publish the paper, after addressing the points below (minor revisions).*
*The language, figures and tables are generally of high quality and easy to read. The structure is easy to follow, and the balance between data, results and discussion is well suited for a publication. However, I recommend improving readability by splitting many long complex sentences into shorter sentences.*

Thanks for the suggestion; we will follow this advice while revising the manuscript.

*One aspect I was missing was an analysis and/or discussion of the general transferability of the results to other parts of the world, especially where terrain or climate conditions differ from the Davos region. One could assume that not only the total size of the study area matters, but rather the size of avalanche terrain area. Some terrain is not able to produce larger avalanches, while other types of terrain produce many large avalanches. Some climates produce many large natural avalanches, while others less so. If the authors could add these aspects to the discussion, it may be easier for the readers or future studies to generalise, add to, or test the results.*

We will discuss this issue in the revised manuscript at the beginning of the Discussion section where we already discuss limitations of our dataset.

*Now follows specific comments, with reference to line numbers in the manuscript:*

*#7 Add size of study area, number of avalanche observations and number of regional danger assessments.*

We will add these numbers to the Abstract.

*#13 Could the sentence ending in "...given day" be improved by adding "an danger level" at the end?*

Yes, thanks, we will change as suggested.

*#15 Could the sentence be improved by replacing "may allow revisiting" by "suggest reworking of"?*

Yes, of course, it makes it a stronger case.

*#18 Add "according to our data" after "km$^2$"*

We will add this restriction.

*#24 Improve the sentence starting with "For these…"*

*We will elaborate this statement in the revised manuscript.*

*#35 Improve flow (order of words) of sentence*

We will consider rephrasing this statement in the revised manuscript.

*#35-38: May also use the EAWS description "Avalanche danger is a function of snowpack stability, its spatial distribution and avalanche size" (https://www.avalanches.org/wp-content/uploads/2019/07/general-assembly-oslo_minutes_EAWS.pdf)*

Thanks for the suggestion. As the description is not really new, we prefer referring to previously published articles.

*#50-51 This description should be updated. The latest EAWS matrix specifically accounts for size (https://www.avalanches.org/standards/eaws-matrix/)*

We refer to recent developments, which are not finished yet and are well described in the references we provide. The version of the Bavarian matrix you refer to is an intermediate step, we may refer to as you wish.

*#75 Explain if the size of the area of avalanche observations is equal to the size of the forecasting region*

The size of the area of avalanche observations corresponds to the typical size of a so-called warning region. The warning region that includes Davos is even a bit smaller, but still representative of the study area.

*#75 Add a short description of how these observations were obtained. Information is provided in the discussion chapter, but it would be logical for the reader to learn about the data upfront. Did the observations cover the entire 360 km$^2$? The entire winter season? All seasons with the same rigorousness?*

We will add some more details on how the observations were made in the revised manuscript.

*#107 Could replace "In other words, on" by "On average,"*

Yes, one could. We will consider rephrasing the start of the sentence in the revised manuscript.

*#109 Add a sentence at the beginning of the paragraph, about why the work described in the paragraph was carried out. E.g., "The forecasting data were scrutinized, in order to adjust danger levels to the most realistic values."*

Thanks for the suggestion. We will add a sentence at the beginning of the paragraph.

*#126 Replacing "Moreover, there were also days, 17 in total" with "This was also the case for 17 days" could improve the readability*

We will improve readability of this sentence.

*#165 Add a descriptor for the values, probably "median values"*

Yes, thanks, we provide indeed the median values. We will specify this in the revised manuscript.

*#175 Replace "for" by "to"*

We will change as suggested.

*#277 Replace "to" by "of"*

We will change as suggested.

*#289 Since the Eckerstorfer el al. study, the number of Sentinel-1 satellites has doubled with 1B in orbit. Thus, the statement of too poor temporal resolution is less valid today. I suggest to add this information to the sentence.*

We have actually checked this statement last fall and also figured out that S1 now consists of two satellites S1-A and S1-B, which alternately image central Europe every six days from the same orbit. Based on this information we considered the temporal resolution in the Alps as still rather poor, i.e. not sufficient for operational forecasting.
We will clarify this point in the revised manuscript.

*#305 Spell out what is meant by "the potential impact"*

We mean that avalanches not causing any damage, or e.g. not reaching the road, are more likely to be not reported.

*#374 Terrain usage probably also decrease from level 2 to 3, as well as from 3 to 4. It would be useful to add references, if these exist, on the differences in terrain usage between danger levels 1, 2, 3 and 4.*

Thanks for this suggestion. We agree that there may be some decrease in usage frequency already from 2–*Moderate* to 3–*Considerable*. We are aware of two studies that looked into that issue. Techel et al. (2015) analysed avalanche risk based on accident data and usage frequency, they inferred usage frequency by exploring two social media mountaineering websites; their study showed a decrease in ski touring activity with regard to danger level (2–*Moderate* vs. 3–*Considerable*). Wäger and Zweifel (2008) also reported a decrease in touring activity, but no change with regard to off-piste skiing. In the region of Davos, ski touring and off-piste skiing are equally relevant.

*#377 Explain in more detail how you arrive at the number 10.*

This is an informed guess. The number may as well be 20. As described in the paper the average number of natural avalanches at danger level 4–*High* was 48. Given the size of our study area,

360 km$^2$, one obtains 13 avalanches per 100 km$^2$, hence we wrote "at least 10". Considering that we do not have full observation coverage in our study area, we could as well suggest about 20 avalanches per 100 km$^2$. This means the term "many" can be quantified, roughly in the range of 10-20, and it becomes clear that "many" is not just one to three.

*#384-389 I find parts of this paragraph unfinished. I would recommend arguing or substantiating why you make the statements "need to" and "should not". I would also recommend to put the sentence "The actual locations…" into context (e.g. the wordings of the NA/CMAH and EADS wrt. spatial distribution).*

Recent discussions in the EAWS and publications (e.g. on the CMAH) have shown that there is potential for confusion with regard to terminology, in particular with regard to avalanche probability. Hence, we simply considered it useful to point out some of the differences since we use some of the terms in the paper as well. There is a similar issue with the spatial distribution of stability as referred to, for instance, in the EAWS definition, you mentioned above. The term spatial distribution can be interpreted in different ways, for instance, spatially and non-spatially. However, the factor contributing to the danger level is the frequency of triggering spots, a non-spatial property. Where the spots in the terrain are located, is not relevant for the definition of the danger levels, only their frequency is relevant. This issue will be clarified in detail in an upcoming manuscript by Techel et al. (2020, in preparation).

---

## Author Comment (AC3) · 15 Jan 2020

**Reply to Reviewer #2**

We thank the reviewer for the positive feedback, insightful comments and constructive suggestions.

*The present manuscript describes an analysis that takes advantage of a large existing dataset of observed and mapped avalanches to explore the relationship between avalanche danger levels and avalanche occurrence in the Davos region of Switzerland. The research is of high quality, and the results contribute valuable insights to the current discussion on avalanche forecasting practices and consistency. The various analyses described in the manuscript offers useful information on the role avalanche size in avalanche forecasting, and the recommendation on the number of expected avalanches at danger rating level High has the potential to be the starting point for making avalanche forecasting more objective by replacing the existing qualitative descriptors in the danger scale with more objective quantitative measures.*
*The manuscript fits well with the mandate of The Cryosphere journal, and it will offer great value for avalanche safety researchers and practitioners. However, despite the obvious strength of the research, I believe that the manuscript has a few weaknesses that should be addressed before the manuscript is published. My concerns mainly relate to the presentation of the dataset correction procedures and the qualitative description of the results. I hope that my comment below are useful for making the manuscript even more impactful.*

*GENERAL COMMENTS AND SUGGESTIONS*
*Correction procedure (Lines 109-140)*
*Given that the objective of your paper is to examine the relationship between avalanche danger ratings and avalanche activity, manually changing danger ratings based on observed avalanche activity prior to analysis seems risky. While I do not necessarily disagree with the approach, I have the following recommendations for making it more transparent for the reader:*

*To put the number of days with corrected the danger ratings into perspective, it would be useful to provide readers with counts and proportions of danger ratings prior to correction right at the beginning of this section. This information is currently only available for the corrected danger ratings (Table 2). Having this information upfront would help readers to understand how much of the dataset was modified.*

We agree and will add the following Table to the revised manuscript.

**Table A1: Frequency of danger levels before and after corrections. Also given are the changes per danger level.**

| Danger level | Number of days before corrections | Change of danger level | | | | | Number of days after corrections |
|---|---|---|---|---|---|---|---|
| | | -2 | -1 | 0 | +1 | +2 | |
| 1–Low | 306 | - | - | 303 | 1 | 2 | 303 |
| 2–Moderate | 1809 | - | 0 | 1765 | 32 | 12 | 1766 |
| 3–Considerable | 1367 | 0 | 0 | 1310 | 57 | | 1366 |
| 4–High | 47 | 0 | 21 | 24 | 2 | - | 94 |
| 5–Very High | 4 | 1 | 1 | 2 | - | - | 4 |

*The description of the correction procedure refers several times to the fact that avalanche activity was 'unusually high' or 'unusually low'. However, you do never explicitly specify what your expectations regarding avalanche activity actually are and how you determined your thresholds (one exception is the recoding of moderate days with AAI > 1.0). Being more explicit about your criteria would make your procedure more transparent.*

We agree and regret the confusion. The procedure and the criteria are as follows:

1. We evaluated all days with danger levels 4–*High* or 5–*Very High* and a value of the AAI ≤ 1 ("zero or unusually low").
2. We evaluated all days with danger level 3–*Considerable* and a value of the AAI > 13.6 (after the first correction step), which was the median AAI for the days with danger level 4–*High* or 5–*Very High* (line 120).
3. We evaluated all days with danger level 2–Moderate and a value of the AAI > 1, which was the median AAI for the days with danger level 3–*Considerable*.
4. We evaluated all days with danger level 1–*Low* and value of the AAI > 1.

*I personally found the description of the correction procedure somewhat difficult to follow due to many details described in the text. I wonder whether a diagram (e.g., flow chart) showing which danger ratings were changed to what and for what reason would help the reader to better understand the magnitude of your changes and their potential impact on the subsequent analysis.*

Thanks for the suggestion. We hope that we can address your concern with Table A shown above and the correction procedure described above.

*The numbers in your description of the changes applied to danger with High and Very High danger ratings (Lines 111-116) do not seem to add up properly.*

Thanks for checking. We will do as well and hopefully the procedure will become easier to follow with the addition of Table A.

*Overall, you changed the danger rating in 122 of 3533 days, which only amounts to 3.5%. This seems like a rather small amount and my initially thought was that the correction procedure was unnecessarily complicated given that it will likely only have a minor impact on the analysis. However, it represents 9% of the days with avalanche activity, and, if I understand your descriptions correct, the number of days with danger ratings High and Very High changed substantially through the correction procedure. The number of days with a High danger rating was first reduced from 44 to 26 (-18) (Line 116) and then increased again to 94 (+68) (Line 141). This means that only 28% of the days with High danger ratings in the analysis dataset were originally assigned a High danger rating. Given that the High danger rating sample plays an important role in the subsequent analysis, I believe that the impact of the correction procedure on the nature of the dataset should be described more clearly.*

The main effect of the correction procedure is that the median AAI for days with 4–*High* increased from about 10 to about 21. For days with 3–*Considerable*, the median AAI was 1 and did not change due to the corrections. Hence, the difference in avalanche activity between 3–*Considerable* and 4–*High* was already very prominent before the correction procedure (before and after the corrections: *U*-test, p < 0.001). With regard to avalanche size, the effects are less prominent. For instance, the finding that size 2 avalanches are the most frequent ones will not change due to the corrections.

*A brief discussion of the potential effect of the correction procedure on the analysis results in the discussion section would further acknowledge its impact. I think it is important to explicitly mention that there is potential for a bit of a circular argument here: You corrected the danger rating levels based on avalanche activity expectations to later analyze exactly this relationship.*

We agree and will add a short paragraph in the Discussion section along the lines described above.

*Description of analysis methods*
*The section titled 'Data and methods' only includes descriptions of the derivation of the avalanche size, the danger rating dataset and the quality control and correction procedure but seems to completely skip a description of the actual analysis approach and statistical methods employed. This seems rather unusual. I believe the manuscript would benefit a short overview of the analysis approach that describes the measures used (e.g., avalanche activity index, proportion of days with avalanches, etc.) and how they relate to the components of avalanche hazard (e.g., snow stability, frequency of locations) in the methods section.*

We agree and will complement the Data and Methods section as suggested.

*Statistical support for qualitative descriptions*
*Much of the description of the observed patterns are rather qualitative with some statistical tests here and there. I am wondering whether some of the statement could be supported with statistical test statistics. I believe that this would considerable strengthen the power of the manuscript.*

We agree and are fully aware that we did not provide many statistical test results. We did so since we believe that the analysis is simple and the data actually speak for itself. However, we will add some more statistical test results in the revised manuscript to better support some of the main findings.

*Description of study area*
*On line 377, you provide recommendations about the number of expected avalanches at danger rating level 4-High (at least 10 per 100 km2), and on line 416, you suggest that the term "many avalanches" should mean on the order of at least about 10 avalanche per 100 km2. I believe that this is an interesting result. However, while you highlight that avalanche occurrence probability depends on scale as it is a combination of stability, its distribution within the forecast area and the size of the forecast area, it seems to me that the nature of the terrain in the forecast region would also have a substantial impact on the suggested number. I therefore wonder whether a more detailed description of the nature of the avalanche terrain in the study area (e.g., number of avalanche paths of different size, total extent of avalanche terrain) would offer valuable context for understanding the results and recommendations.*

We agree that the type of terrain certainly affects avalanche activity. We will provide the proportion of avalanche terrain in the revised manuscript.

*Insight into avalanche warning practices*
*In several places in this manuscript, you comment on the somewhat unexpected differences in the observed numbers of dry and wet avalanches at the same danger rating level (e.g., Lines 323-324, Lines 375-376). However, there is no explicit statement in the discussion or conclusion section that points out that these observations might indicate inconsistencies in forecasting practices. I think that a statement like this would fit nicely with the recent literature on avalanche forecasting inconsistency and further contribute to this research.*

Thanks for that suggestion; we do actually mention it in the Conclusions (line 414), but will check whether we can include it in the Discussion section as well.

*LINE-SPECIFIC COMMENTS*
*1. **Line 95 – Figure 1** The exponential increase presented in Figure 1 seems to be the direct consequence of the classification criteria presented in Table 1. I wonder whether plotting the log of avalanche area versus avalanche size class would be more useful to highlight that the approach*

*classifies avalanche in the spirit of the Canadian size classification. The number of avalanches per class shown in the chart do not add up to the total number of avalanches given in the caption.*

We have deliberately chosen the length, as this measure is the one most practitioners in Europe can well relate to. Also, the length is included in the EAWS definition of avalanche sizes. In addition, we provide the median area per size class in Table 1.

Thanks for checking the numbers. There is indeed an error since for some of the very small avalanches a meaningful value of length could not be derived so that the total number reduces to 13,802.

*2. **Line 100** It might be useful to explicitly state that the weight of 0.81 is appropriate because it is highly likely that the avalanches without known triggers were likely natural avalanches.*

We will state that explicitly.

*3. **Line 191 – Table 3** It might be useful to add row percentages to the columns to better highlight the relationship between avalanche size distribution and danger rating.*

Thanks for the suggestion. We will consider adding the percentage values; by the way, they are already shown in Figure 3.

*4. **Line 192** The Kruskal-Wallis test only indicates whether there are any differences in the avalanche size distributions among all danger rating levels. You could follow-up with pairwise Wilcoxon rank-sum tests between adjacent danger rating levels to determine where exactly the differences are.*

Thanks for the suggestion. We have actually done so and considered all pairwise combinations, but missed to report the results. We will do so in the revised manuscript.

*5. **Line 196 – Figure 3** I think it would be best if the proportion scales in all charts would range from 0 to 1 and be styled the same.*

We agree and change the scale on the y-axis in Figure 3a.

*6. **Line 196 – Figure 3c** If I understand your analysis correctly, Fig. 3c depicts average or median number of avalanches of different sizes per day at different avalanche danger levels. A bar chart does not seem an appropriate way to display this information as bar charts are typically used to depict proportions. Populating the same layout grid with a series of box plots instead of vertical bars would represent not only the magnitude difference in number of avalanches between danger ratings and avalanche size, but also the range within the observations.*

Thanks for your suggestion. We prepared the box plots for the number of avalanches, but realize it would not be possible to provide all data in one graph as you suggest. Moreover, the information would be hard to depict since the scale on the y-axis would range from 0 to 250. Therefore, we prefer to keep Figure 3c as presented.

*7. **Line 209** Can the statement "The median as well as the 90-percentile avalanche length did not increase for the danger levels 1-Low to 4-High." be substantiated with a statistical test result?*

We will provide more details on the median and 90 percentile (see Figure below) The median length was 164, 154, 163 and 198 m for the danger levels 1–*Low* to 4–*High*, respectively. The corresponding 90 percentile lengths are: 527, 449, 448, 566 m.

[Figure]

**Figure A1: Avalanche length per avalanche danger level (N=13'745).**

*8. **Line 217 and Figure 4a** It seems odd to combine avalanches with unknown snow conditions with the mixed category in the snow conditions analysis (Section 3.3) as these are very different categories. I think it would be better to leave these avalanches out of the analysis all together.*

Based on our experience the unknown conditions are often related to mixed conditions.

*9. **Line 227 and Figure 4d** The difference in the number of avalanches per day between dry and wet avalanches under avalanche danger level 1-Low shown in Fig. 4d seems minute. Can this statement be supported with a statistical test?*

As stated in lines 226-229 there were distinct differences between days with wet-snow avalanches and days with dry-snow avalanche on days when the danger level was 1–*Low* – though the number of cases is rather small:
On the 10 days with natural wet-snow avalanches, the number of avalanches was 18 and the total AAI was 15.2. Whereas on the 6 days with natural dry-snow avalanches, the number of avalanches was 8 and the total AAI was 0.71. As stated, the AAI was more than ten times larger.
On the other hand, in Figure 4d the average number of avalanches per day is shown, which is 1.3 for dry-snow and 1.8 for wet-snow conditions; this difference is not particularly large as you point out and statistically not significant.

*10. **Line 233** I assume that this discussion should be referring to the AVERAGE or MEDIAN number of avalanches per day.*

We will specify and add average in the revised manuscript.

*11. **Line 241 – Figure 4d** Same comment as for Figure 3c*

Please see our response to point #6 above.

*12. **Line 253** The statement "… size 4 avalanches were five times more frequent among the natural than the human-triggered avalanche." does not seem to be supported by Figure 5a.*

We agree that our statement was not adequate since it refers to the absolute number of size 4 avalanches. However, as there are three times more natural than human-triggered avalanches the proportions are 2.9 % vs. 1.6 % , still a significant difference (proportion test, p = 0.02). We will re-word the statement in the revised manuscript.

*13. **Line 256 – Figure 5c** Same comment as for Figure 3c and Figure 4d*

Please see our response to point #6 above.

*14. **Lines 268-273** Same comment as earlier regarding the average/median number of avalanches per day.*

Thanks for pointing this out, we will specify that the number refers to the average.

*15. **Line 293** I think it would be useful for the reader if the fact that no temporal trends in the avalanche size distribution were detected in the analysis dataset was included and substantiated in the initial description of the dataset.*

We may reconsider where to add this statement. We think it can be in either of the two sections.

*16. **Lines 316-324** It seems to me that some of the explanations of the data correction procedure described in this paragraph should be included in the methods section.*

As we will provide more details in the Methods section on the correction procedure (see above), we will revisit this paragraph and check where the explanations are needed.

*17. **Lines 340-346** In this section, you refer to the avalanche size distribution of human triggered avalanches, and the topic comes up again on Line 407. However, I did not find this explicit analysis in your manuscript. If I read your manuscript correctly, you only analyzed the number of human triggered avalanches under different danger ratings but not their size distributions. It seems to me that such an analysis would nicely complement your existing analyses.*

Harvey (2002) considered avalanches that caused damage to either people, infrastructure or forest, and reported avalanche size. Hence, we discuss his study to compare his results to our findings.
In fact, we have analysed the size distribution for human-triggered avalanches at the different danger levels. The distributions are very similar to the overall distribution shown in Figure 5a. For instance, at danger level 3–*Considerable* and 4–*High*, the proportion of size 3 avalanches is 0.74 and 0.73 (proportion test, p = 0.77). For size 4 avalanches the corresponding proportions are 0.14 and 0.19 (proportion test, p = 0.36). We will consider adding this information to the Results section.

*18. **Lines 355** Same comment as earlier regarding the average/median number of avalanches per day.*

Thanks, we will specify the number as the average in the revised manuscript.

*19. **Lines 378-381** The description of the results of Bründl et al (2019) is a bit confusing to me. What are the five frequency classes and how do they relate to the results presented in this paper?*

We will clarify the approach presented in Bründl et al. (2019).

*20. **Line 389** Add "and terrain choices." at the end of this sentence*

Thanks for the suggestion, we will add this additional factor.

---

## Author Response (AR1)

**Reply to Reviewer #1**

We thank the reviewer for the positive feedback and his constructive suggestions.

*The paper is of high scientific quality. It brings forward important quantitative results on the occurrence of snow avalanches in the Eastern Alps, and provides an important analysis and discussion of the implications for assessment and forecasting of avalanche danger. The study uses a unique and rich data set on observed avalanches and regional avalanche danger estimates, in order to reduce a critical knowledge gap that has limited the development of objective procedures for determining the avalanche danger level. The publication of the paper will contribute to improved workflows, standards and eventually better avalanche forecasting products in the future.*
*My recommendation to the editors is to publish the paper, after addressing the points below (minor revisions).*
*The language, figures and tables are generally of high quality and easy to read. The structure is easy to follow, and the balance between data, results and discussion is well suited for a publication. However, I recommend improving readability by splitting many long complex sentences into shorter sentences.*

Thanks for the suggestion; we followed this advice while revising the manuscript.

*One aspect I was missing was an analysis and/or discussion of the general transferability of the results to other parts of the world, especially where terrain or climate conditions differ from the Davos region. One could assume that not only the total size of the study area matters, but rather the size of avalanche terrain area. Some terrain is not able to produce larger avalanches, while other types of terrain produce many large avalanches. Some climates produce many large natural avalanches, while others less so. If the authors could add these aspects to the discussion, it may be easier for the readers or future studies to generalise, add to, or test the results.*

We added the proportion of avalanche terrain (67%) based on the classification by Harvey et al. (2018). Moreover, the snow climate can be described as transitional (McClung and Schaerer, 2006). As you indicate terrain and snow climate influence avalanche activity. However, we deem it speculative to predict the characteristics of avalanche activity in regions with other snow climates or terrain characteristics. More similar studies are needed. We do refer to some studies from North America and the New Zealand in the Discussion section. We suppose the avalanche size distribution to be fairly independent of terrain and climate characteristics. However, the frequency and intensity of occurrence will certainly vary.
We now mention this issue in the Discussion section (lines 374-377).

*Now follows specific comments, with reference to line numbers in the manuscript:*

*#7 Add size of study area, number of avalanche observations and number of regional danger assessments.*

We added these numbers to the Abstract (lines 7-8).

*#13 Could the sentence ending in "...given day" be improved by adding "an danger level" at the end?*

Changed as suggested (line 13).

*#15 Could the sentence be improved by replacing "may allow revisiting" by "suggest reworking of"?*

Changed as suggested (line 16).

**Reply to Reviewer #2**

We thank the reviewer for the positive feedback, insightful comments and constructive suggestions.

*The present manuscript describes an analysis that takes advantage of a large existing dataset of observed and mapped avalanches to explore the relationship between avalanche danger levels and avalanche occurrence in the Davos region of Switzerland. The research is of high quality, and the results contribute valuable insights to the current discussion on avalanche forecasting practices and consistency. The various analyses described in the manuscript offers useful information on the role avalanche size in avalanche forecasting, and the recommendation on the number of expected avalanches at danger rating level High has the potential to be the starting point for making avalanche forecasting more objective by replacing the existing qualitative descriptors in the danger scale with more objective quantitative measures.*
*The manuscript fits well with the mandate of The Cryosphere journal, and it will offer great value for avalanche safety researchers and practitioners. However, despite the obvious strength of the research, I believe that the manuscript has a few weaknesses that should be addressed before the manuscript is published. My concerns mainly relate to the presentation of the dataset correction procedures and the qualitative description of the results. I hope that my comment below are useful for making the manuscript even more impactful.*

*GENERAL COMMENTS AND SUGGESTIONS*
*Correction procedure (Lines 109-140)*
*Given that the objective of your paper is to examine the relationship between avalanche danger ratings and avalanche activity, manually changing danger ratings based on observed avalanche activity prior to analysis seems risky. While I do not necessarily disagree with the approach, I have the following recommendations for making it more transparent for the reader:*

*To put the number of days with corrected the danger ratings into perspective, it would be useful to provide readers with counts and proportions of danger ratings prior to correction right at the beginning of this section. This information is currently only available for the corrected danger ratings (Table 2). Having this information upfront would help readers to understand how much of the dataset was modified.*

We agree and added the following Table as supplement to the revised manuscript.

**Table A: Frequency of danger levels before and after corrections. Also given are the changes per danger level.**

| Danger level | Number of days before corrections | Change of danger level | | | | | Number of days after corrections |
|---|---|---|---|---|---|---|---|
| | | -2 | -1 | 0 | +1 | +2 | |
| 1–Low | 306 | - | - | 303 | 1 | 2 | 303 |
| 2–Moderate | 1809 | - | 0 | 1765 | 32 | 12 | 1766 |
| 3–Considerable | 1367 | 0 | 0 | 1310 | 57 | | 1366 |
| 4–High | 47 | 0 | 21 | 24 | 2 | - | 94 |
| 5–Very High | 4 | 1 | 1 | 2 | - | - | 4 |

*The description of the correction procedure refers several times to the fact that avalanche activity was 'unusually high' or 'unusually low'. However, you do never explicitly specify what your expectations regarding avalanche activity actually are and how you determined your thresholds (one exception is the recoding of moderate days with AAI > 1.0). Being more explicit about your criteria would make your procedure more transparent.*

We agree and regret the confusion. The procedure and the criteria are as follows:

1. We evaluated all days with danger levels *4–High* or *5–Very High* and a value of the AAI ≤ 1 ("zero or unusually low").
2. We evaluated all days with danger level *3–Considerable* and a value of the AAI > 13.6 (after the first correction step), which was the median AAI for the days with danger level *4–High* or *5–Very High* (line 120).
3. We evaluated all days with danger level *2–Moderate* and a value of the AAI > 1, which was the median AAI for the days with danger level *3–Considerable.*
4. We evaluated all days with danger level *1–Low* and value of the AAI > 1.

*I personally found the description of the correction procedure somewhat difficult to follow due to many details described in the text. I wonder whether a diagram (e.g., flow chart) showing which danger ratings were changed to what and for what reason would help the reader to better understand the magnitude of your changes and their potential impact on the subsequent analysis.*

Thanks for the suggestion. We hope that we can address your concern with Table A shown above and the correction procedure described above, which we now both provide in the Supplement.

*The numbers in your description of the changes applied to danger with High and Very High danger ratings (Lines 111-116) do not seem to add up properly.*

Thanks for checking. We checked and the numbers should now be consistent with Table A in the Supplement.

*Overall, you changed the danger rating in 122 of 3533 days, which only amounts to 3.5%. This seems like a rather small amount and my initially thought was that the correction procedure was unnecessarily complicated given that it will likely only have a minor impact on the analysis. However, it represents 9% of the days with avalanche activity, and, if I understand your descriptions correct, the number of days with danger ratings High and Very High changed substantially through the correction procedure. The number of days with a High danger rating was first reduced from 44 to 26 (-18) (Line 116) and then increased again to 94 (+68) (Line 141). This means that only 28% of the days with High danger ratings in the analysis dataset were originally assigned a High danger rating. Given that the High danger rating sample plays an important role in the subsequent analysis, I believe that the impact of the correction procedure on the nature of the dataset should be described more clearly.*

The main effect of the correction procedure is that the median AAI for days with *4–High* increased from about 10 to about 21. For days with *3–Considerable*, the median AAI was 1 and did not change due to the corrections. Hence, the difference in avalanche activity between *3–Considerable* and *4–High* was already very prominent before the correction procedure (before and after the corrections: *U*-test, $p < 0.001$). With regard to avalanche size, the effects are less prominent. Size 2 avalanches were the most frequent ones at danger levels *1–Low* to *4–High* before and after the corrections.
We added a short paragraph on the impact of the correction procedure in the Discussion section (lines 398-404).

*A brief discussion of the potential effect of the correction procedure on the analysis results in the discussion section would further acknowledge its impact. I think it is important to explicitly mention that there is potential for a bit of a circular argument here: You corrected the danger rating levels based on avalanche activity expectations to later analyze exactly this relationship.*

We agree, see above, and added a short paragraph in the Discussion section along the lines described above (lines 398-404).

***Description of analysis methods***
*The section titled 'Data and methods' only includes descriptions of the derivation of the avalanche size, the danger rating dataset and the quality control and correction procedure but seems to completely skip a description of the actual analysis approach and statistical methods employed. This seems rather unusual. I believe the manuscript would benefit a short overview of the analysis approach that describes the measures used (e.g., avalanche activity index, proportion of days with avalanches, etc.) and how they relate to the components of avalanche hazard (e.g., snow stability, frequency of locations) in the methods section.*

We now describe the analysis methods in more detail (lines 163-173).

***Statistical support for qualitative descriptions***
*Much of the description of the observed patterns are rather qualitative with some statistical tests here and there. I am wondering whether some of the statement could be supported with statistical test statistics. I believe that this would considerable strengthen the power of the manuscript.*

We agree and are fully aware that we did not provide many statistical test results. We did so since we believe that the analysis is simple and the data actually speak for themselves.
We now provide more statistical test results to better support some of the main findings (e.g. lines 178, 231, 237).

***Description of study area***
*On line 377, you provide recommendations about the number of expected avalanches at danger rating level 4-High (at least 10 per 100 km2), and on line 416, you suggest that the term "many avalanches" should mean on the order of at least about 10 avalanche per 100 km2. I believe that this is an interesting result. However, while you highlight that avalanche occurrence probability depends on scale as it is a combination of stability, its distribution within the forecast area and the size of the forecast area, it seems to me that the nature of the terrain in the forecast region would also have a substantial impact on the suggested number. I therefore wonder whether a more detailed description of the nature of the avalanche terrain in the study area (e.g., number of avalanche paths of different size, total extent of avalanche terrain) would offer valuable context for understanding the results and recommendations.*

We agree that the type of terrain certainly affects avalanche activity. We now provide the proportion of avalanche terrain (lines 79-81).

***Insight into avalanche warning practices***
*In several places in this manuscript, you comment on the somewhat unexpected differences in the observed numbers of dry and wet avalanches at the same danger rating level (e.g., Lines 323-324, Lines 375-376). However, there is no explicit statement in the discussion or conclusion section that points out that these observations might indicate inconsistencies in forecasting practices. I think that a statement like this would fit nicely with the recent literature on avalanche forecasting inconsistency and further contribute to this research.*

Thanks for that suggestion; we do actually mention it in the Conclusions (line 500). We also added this point to the Discussion section (lines 395-397).

***LINE-SPECIFIC COMMENTS***
*1. **Line 95 – Figure 1** The exponential increase presented in Figure 1 seems to be the direct consequence of the classification criteria presented in Table 1. I wonder whether plotting the log of avalanche area versus avalanche size class would be more useful to highlight that the approach*

*classifies avalanche in the spirit of the Canadian size classification. The number of avalanches per class shown in the chart do not add up to the total number of avalanches given in the caption.*

We have deliberately chosen the length, as this measure is the one most practitioners in Europe can well relate to. Also, the length is included in the EAWS definition of avalanche sizes. In addition, we provide the median area per size class in Table 1.
Thanks for checking the numbers. There is indeed an error since for some of the very small avalanches a meaningful value of length could not be derived so that the total number reduces to 13,802.
We now mention this in lines 89-91. Moreover, we now provide the number of cases in Table 1.

*2. **Line 100** It might be useful to explicitly state that the weight of 0.81 is appropriate because it is highly likely that the avalanches without known triggers were likely natural avalanches.*

We now explicitly state that the value is appropriate (line 113).

*3. **Line 191 – Table 3** It might be useful to add row percentages to the columns to better highlight the relationship between avalanche size distribution and danger rating.*

Thanks for the suggestion. We added the proportions in Table 2.

*4. **Line 192** The Kruskal-Wallis test only indicates whether there are any differences in the avalanche size distributions among all danger rating levels. You could follow-up with pairwise Wilcoxon rank-sum tests between adjacent danger rating levels to determine where exactly the differences are.*

Thanks for the suggestion. We now describe some of the pairwise comparisons (lines 229-232) and also provide all *p*-values in Table B of the Supplement.

*5. **Line 196 – Figure 3** I think it would be best if the proportion scales in all charts would range from 0 to 1 and be styled the same.*

We agree and changed the scale on the y-axis in Figure 3a.

*6. **Line 196 – Figure 3c** If I understand your analysis correctly, Fig. 3c depicts average or median number of avalanches of different sizes per day at different avalanche danger levels. A bar chart does not seem an appropriate way to display this information as bar charts are typically used to depict proportions. Populating the same layout grid with a series of box plots instead of vertical bars would represent not only the magnitude difference in number of avalanches between danger ratings and avalanche size, but also the range within the observations.*

Thanks for your suggestion. We prepared the box plots for the number of avalanches, but realize it would not be possible to provide all data in one graph as you suggest. Moreover, the information would be hard to depict since the scale on the y-axis would range from 0 to 250. Therefore, we prefer to keep Figure 3c as presented.

*7. **Line 209** Can the statement "The median as well as the 90-percentile avalanche length did not increase for the danger levels 1-Low to 4-High." be substantiated with a statistical test result?*

We now provide more details on the median avalanche length (lines 247-250). Also see Figure A below, which we included into the Supplement.

[Figure]

**Figure A: Avalanche length per avalanche danger level (N=13'745).**

*8. **Line 217 and Figure 4a** It seems odd to combine avalanches with unknown snow conditions with the mixed category in the snow conditions analysis (Section 3.3) as these are very different categories. I think it would be better to leave these avalanches out of the analysis all together.*

Based on our experience the unknown conditions are often related to mixed conditions. The size distribution for unknown and mixed were statistically not different (*U*-test, $p = 0.82$).
We added this information (lines 257-258).

*9. **Line 227 and Figure 4d** The difference in the number of avalanches per day between dry and wet avalanches under avalanche danger level 1-Low shown in Fig. 4d seems minute. Can this statement be supported with a statistical test?*

As stated in lines 226-229 there were distinct differences between days with wet-snow avalanches and days with dry-snow avalanche on days when the danger level was 1–*Low* – though the number of cases is rather small:
On the 10 days with natural wet-snow avalanches, the number of avalanches was 18 and the total AAI was 15.2. Whereas on the 6 days with natural dry-snow avalanches, the number of avalanches was 8 and the total AAI was 0.71. As stated, the AAI was more than ten times larger.
On the other hand, in Figure 4d the average number of avalanches per day is shown, which is 1.3 for dry-snow and 1.8 for wet-snow conditions; this difference is not particularly large as you point out and statistically not significant.
We now provide more details and rephrased that paragraph (lines 274-279).

*10. **Line 233** I assume that this discussion should be referring to the AVERAGE or MEDIAN number of avalanches per day.*

We added the average (line 284).

*11. **Line 241 – Figure 4d** Same comment as for Figure 3c*

Please see our response to point #6 above. At 4–*High* the maximum number of avalanches per day was 255 so that the y-scale would need to range from 0 to 300 and not much could be seen at the lower danger levels.

*12. **Line 253** The statement "... size 4 avalanches were five times more frequent among the natural than the human-triggered avalanche." does not seem to be supported by Figure 5a.*

We agree that our statement was not correct since it refers to the absolute number of size 4 avalanches. Since there are three times more natural than human-triggered avalanches, the proportions are 2.9 % vs. 1.6 %, still a significant difference (proportion test, $p = 0.02$). We corrected the statement (lines 306-307).

*13. **Line 256 – Figure 5c** Same comment as for Figure 3c and Figure 4d*

Please see our response to point #6 above.

*14. **Lines 268-273** Same comment as earlier regarding the average/median number of avalanches per day.*

We added the average (line 329).

*15. **Line 293** I think it would be useful for the reader if the fact that no temporal trends in the avalanche size distribution were detected in the analysis dataset was included and substantiated in the initial description of the dataset.*

We now mention that fewer avalanches were recorded in the first six years of the study period in the Data and Methods section (lines 84-85). Also, we added some more details in the Discussion section (lines 355-359).

*16. **Lines 316-324** It seems to me that some of the explanations of the data correction procedure described in this paragraph should be included in the methods section.*

We now provide more details in the Methods section on the correction procedure (see above) and also added a paragraph to discuss possible effects of the corrections (lines 398-404).

*17. **Lines 340-346** In this section, you refer to the avalanche size distribution of human triggered avalanches, and the topic comes up again on Line 407. However, I did not find this explicit analysis in your manuscript. If I read your manuscript correctly, you only analyzed the number of human triggered avalanches under different danger ratings but not their size distributions. It seems to me that such an analysis would nicely complement your existing analyses.*

Harvey (2002) considered avalanches that caused damage to either people, infrastructure or forest, and reported avalanche size. Hence, we discuss his study to compare his results to our findings.
In fact, we have analysed the size distribution for human-triggered avalanches at the different danger levels. The distributions are very similar to the overall distribution shown in Figure 5a. For instance, at danger level *3–Considerable* and *4–High*, the proportion of size 3 avalanches is 0.74 and 0.73 (proportion test, $p = 0.77$). For size 4 avalanches the corresponding proportions are 0.14 and 0.19 (proportion test, $p = 0.36$). Overall avalanche size did not increase with increasing danger level. We added this information to section 3.4 of the Results (lines 315-320).

*18. **Lines 355** Same comment as earlier regarding the average/median number of avalanches per day.*

Thanks, we now specify that the number is the average (line 437).

*19. **Lines 378-381** The description of the results of Bründl et al (2019) is a bit confusing to me. What are the five frequency classes and how do they relate to the results presented in this paper?*

We regret the confusion. In fact, the frequency classes by Bründl et al. (2019) are not relevant in our context. They counted the number of size 4 avalanches per 250 km$^2$ and provided the results in five classes of varying avalanche density. In the lowest class they reported <29 avalanches, whereas the number of avalanche per 250 km$^2$ in the highest class varied between 122 and 202 avalanches. We simply provide the range from the lowest to the highest class: < 29 to 202.
We simplified the description (lines 465-468).

*20. **Line 389** Add "and terrain choices." at the end of this sentence*

Thanks for the suggestion, we added this additional factor (line 477).

*#18 Add "according to our data" after "km$^2$"*

Re-worded as suggested (line 18).

*#24 Improve the sentence starting with "For these…"*

We revised this statement (lines 25-27).

*#35 Improve flow (order of words) of sentence*

We rephrased this sentence (lines 36-38).

*#35-38: May also use the EAWS description "Avalanche danger is a function of snowpack stability, its spatial distribution and avalanche size" (https://www.avalanches.org/wp-content/uploads/2019/07/general-assembly-oslo_minutes_EAWS.pdf)*

Thanks for the suggestion. As the description is not really new, we prefer referring to previously published articles.

*#50-51 This description should be updated. The latest EAWS matrix specifically accounts for size (https://www.avalanches.org/standards/eaws-matrix/)*

We now mention the present version of the Bavarian matrix (line 57).

*#75 Explain if the size of the area of avalanche observations is equal to the size of the forecasting region*

The size of the area of avalanche observations corresponds to the typical size of a so-called warning region. The warning region that includes Davos is even a bit smaller, but still representative of the study area. We now specify that the warning region of Davos is slightly smaller but still in the centre of the study region (lines 116-117).

*#75 Add a short description of how these observations were obtained. Information is provided in the discussion chapter, but it would be logical for the reader to learn about the data upfront. Did the observations cover the entire 360 km$^2$? The entire winter season? All seasons with the same rigorousness?*

We added some more details on how the observations were made (lines 83-85).

*#107 Could replace "In other words, on" by "On average,"*

We reworded the start of the sentence (line 120).

*#109 Add a sentence at the beginning of the paragraph, about why the work described in the paragraph was carried out. E.g., "The forecasting data were scrutinized, in order to adjust danger levels to the most realistic values."*

Thanks for the suggestion. We re-wrote parts of this paragraph (lines 122-129).

*#126 Replacing "Moreover, there were also days, 17 in total" with "This was also the case for 17 days" could improve the readability*

Changed as suggested (line 147).

*#165 Add a descriptor for the values, probably "median values"*

Yes, thanks, we added median values (line 195).

*#175 Replace "for" by "to"*

Changed as suggested (line 205).

*#277 Replace "to" by "of"*

We changed as suggested (line 338).

*#289 Since the Eckerstorfer el al. study, the number of Sentinel-1 satellites has doubled with 1B in orbit. Thus, the statement of too poor temporal resolution is less valid today. I suggest to add this information to the sentence.*

We have actually checked this statement last fall and also figured out that S1 now consists of two satellites S1-A and S1-B, which alternately image central Europe every six days from the same orbit. Based on this information we considered the temporal resolution in the Alps as still rather poor, i.e. not sufficient for operational forecasting.
We clarified this point (lines 349-351).

*#305 Spell out what is meant by "the potential impact"*

We mean that avalanches not causing any damage, or e.g. not reaching the road, are more likely to be not reported.
We now mention this (lines 371-372).

*#374 Terrain usage probably also decrease from level 2 to 3, as well as from 3 to 4. It would be useful to add references, if these exist, on the differences in terrain usage between danger levels 1, 2, 3 and 4.*

Thanks for this suggestion. We agree that there may be some decrease in usage frequency already from *2–Moderate* to *3–Considerable*. We are aware of two studies that looked into that issue. Techel et al. (2015) analysed avalanche risk based on accident data and usage frequency, they inferred usage frequency by exploring two social media mountaineering websites; their study showed a decrease in

ski touring activity with regard to danger level (*2–Moderate* vs. *3–Considerable*). Wäger and Zweifel (2008) also reported a decrease in touring activity, but no change with regard to off-piste skiing. In the region of Davos, ski touring and off-piste skiing are equally relevant.

We now refer to these two study on usage frequency in relation to the avalanche danger level (lines 456-459).

*#377 Explain in more detail how you arrive at the number 10.*

This is an informed guess. As described in the paper the average number of natural avalanches at danger level *4–High* was 48. Given the size of our study area (300 km$^2$), one obtains about 16 avalanches per 100 km$^2$, hence we wrote "at least 10". Considering that we do not have full observation coverage in our study area, we could as well suggest about 20 avalanches per 100 km$^2$. This means the term "many" can be quantified, roughly in the range of 10-20, and it becomes clear that "many" is not just one to three.

We now provide some more background on this estimate (lines 462-464).

*#384-389 I find parts of this paragraph unfinished. I would recommend arguing or substantiating why you make the statements "need to" and "should not". I would also recommend to put the sentence "The actual locations…" into context (e.g. the wordings of the NA/CMAH and EADS wrt. spatial distribution).*

Recent discussions in the EAWS and publications (e.g. on the CMAH) have shown that there is potential for confusion with regard to terminology, in particular with regard to avalanche probability. Hence, we simply considered it useful to point out some of the differences since we use some of the terms in the paper as well. There is a similar issue with the spatial distribution of stability as referred to, for instance, in the EAWS definition, you mentioned above. The term spatial distribution can be interpreted in different ways, for instance, spatially and non-spatially. However, the factor contributing to the danger level is the frequency of triggering spots, a non-spatial property. Where the spots in the terrain are located, is not relevant for the definition of the danger levels, only their frequency is relevant. This issue will be clarified in detail in an upcoming manuscript by (Techel et al., 2020, in preparation).

We have partly edited this paragraph (lines 469-477).

**References**

[revised manuscript text omitted]

---

## Author Response (AR2)

Dear Editor

Many thanks for your prompt and careful read of our manuscript. We much appreciate. We made all the corrections you suggested.

*line 114*
*AAI has not been defined yet. This abbreviation seems to be explained and referenced only 2 pages later (line 163).*

We now introduce the AAI in line 108.

*line 178*
*Spearman rank order correlation->*
*Spearman rank-order correlation*

We introduced $r_s$ for the correlation coefficient.

*line 178 and many other similar places*
*{p} should be in italic*

Changed as suggested.

*line 180*
*well known analysis*
*->well-known analysis*

Changed as suggested.

*line 258*
*was->were*
Changed as suggested.

*line 320*
*Spearman rank correlation->*
*Spearman rank-order correlation?*

See above.

*line 323*
*In total there were ->*
*In total, there were*

Changed as suggested.

*line 435*
*Just a note on the observation bias (in regard to low frequency of size 1 avalanches). Actually, this suggestion is also in line with magnitude-frequency relation of most earthquake catalogs, which miss smaller events. For dealing with this statistical seismologists introduced a so-called magnitude of completeness (Mc), which is the minimum magnitude above which all events are reliably recorded in some area. Usually Mc is obvious to an eye on a cumulative curve of events (the cumulative number of N events greater than a given magnitude) and appears as a departure from the slope.*

Thanks for pointing this out. We added two sentences at the end of the paragraph referring to the similarity with earthquake catalogues (lines 436-438).

*line 458*
*(Wäger and Zweifel, 2008)*
*->*
*Wäger and Zweifel (2008)*

Changed as suggested.

*rom->from*

Changed as suggested.

*line 501*
*reflecting inconsistence usage of the danger scale ->*
*... inconsistent usage ...*

Changed as suggested.

*[Supplement]*
*In point 2, a reference to "(line 120)" is not clear. Can be removed?*

Yes, we deleted the reference to that line in the manuscript; it was a remnant from the response.

[revised manuscript text omitted]